

# Coupled mesoscale-LES modelling of air quality in a polluted city using WRF-LES-Chem

Yuting Wang[1], Yong-Feng Ma[2], Domingo Muñoz-Esparza[3], Jianing Dai[4], Cathy W. Y. Li[4], Pablo Lichtig[4], Roy C.W. Tsang[5], Chun-Ho Liu[6], Tao Wang[1], and Guy P. Brasseur[1,4,7]

[1]Department of Civil and Environmental Engineering, the Hong Kong Polytechnic University, Hung Hom, Kowloon, Hong Kong.
[2]Department of Mechanics & Aerospace Engineering, Southern University of Science and Technology, Shenzhen, 518055, China.
[3]Research Applications Laboratory, National Center for Atmospheric Research, Boulder, CO 80301, USA.
[4]Max Planck Institute for Meteorology, 20146 Hamburg, Germany.
[5]Environmental Protection Department, Hong Kong
[6]Department of Mechanical Engineering, The University of Hong Kong, Hong Kong
[7]Atmospheric Chemistry Observation & Modeling Laboratory, National Center for Atmospheric Research, Boulder, CO 80301, USA.

*Corresponding author:* Yuting Wang (yuting.wang@polyu.edu.hk) and Yong-Feng Ma (mayf3@sustech.edu.cn)

**Abstract.** To perform realistic high-resolution air quality modeling in a polluted urban area, the WRF (Weather Research and Forecasting) model is used with an embedded large-eddy simulation (LES) module and with online chemistry. As an illustration, a numerical experiment is conducted in the megacity Hong Kong, which is characterized by multi-type pollution sources as well as complex topography. The multi-resolution simulations from mesoscale to LES scales are evaluated by comparing to ozone sounding profiles and surface observations. The comparisons show that both mesoscale and LES simulations reproduce well the mean concentrations of the chemical species and their diurnal variations at the background stations. However, the mesoscale simulations largely underestimate the $NO_X$ concentrations and overestimate $O_3$ at the roadside stations due to the coarse representation of the traffic emissions. The LES simulations improve the agreement with the measurements near the road traffic, and the LES with the highest spatial resolution (33.3 m) provides the best results. The LES simulations show more detailed structures in the spatial distributions of chemical species than the mesoscale simulations, highlighting the capability of LES to resolve high-resolution photochemical transformations in urban areas. Compared to the mesoscale model results, the LES simulations show similar evolutions in the profiles of the chemical species as a function of the boundary layer development over a diurnal cycle.

## 1 Introduction

Air pollution represents one of the most important factors affecting human health (WHO, 2013). Human exposure to outdoor air pollution often occurs at street level in urban canyons, and high-resolution air quality forecasting is therefore required to take appropriate preventive actions and limit air pollution (Barzyk et al., 2015; Sauer and Muñoz-Esparza, 2020). Air quality



at street level is difficult to simulate, especially if the model resolution is too coarse to properly resolve urban flows. The complexities in the distribution and variation of the chemical species in the urban canopy are due to several factors. First, the

heterogeneous emissions, such as vehicle emissions at street level, are difficult to represent in coarse models. Second, the dispersion of chemical species in turbulent flows cannot be adequately resolved by mesoscale models. Additionally, the nonlinear turbulence-chemistry interactions in the turbulent planetary boundary layer (PBL), which can be important under polluted situations, are ignored in the coarse models (Li et al., 2021; Wang et al., 2021, 2022).

Because air pollution is determined by both local dynamical circulations and long-distance transport, the accurate simulation

of the species' distribution requires that mesoscale and microscale processes be coupled. Regional models, which adopt global or coarser regional model output as initial and boundary conditions, are usually applied at the spatial resolution of several kilometers, with the effects of turbulent motions being parameterized through commonly used one-dimensional PBL schemes. Mesoscale models can therefore reproduce regional-scale air pollution, but are not adequate for simulating the distribution of physical and chemical variables at the urban neighborhood scale, which requires that the model resolution be less than 1 km.

Moreover, the representation of air pollution near road traffic requires further increasing of the model resolution to be smaller than 40 m (Batterman et al., 2014). In order to explicitly resolve buildings in the models to fully account for the urban effects, the grid spacing should further be increased to at least 10 m (Maronga et al., 2019).

Different downscaling methods are used to represent detailed air pollution distributions in urban or industrial areas. Due to their high computational efficiency, models based on a Gaussian-type redistribution of the wind flow (Forehead and Huynh,

2018) are often adopted to represent chemical species in street canyons. However, these types of models do not directly calculate dynamical and thermal processes and ignore therefore turbulence-chemistry interactions. To better predict the nonlinear photochemical reactions in the PBL over complex underlying surfaces, the turbulent eddies need to be explicitly resolved.

There are three main basic approaches can be used to account for the effects of turbulence on the chemical species: the

Reynolds-averaged Navier-Stokes (RANS) formulation, the Large-Eddy Simulation (LES) approach, and the Direct Numerical Simulation (DNS) method. Among these models, RANS requires the least computational resources, but can only resolve the mean flows and averages out the turbulent fluctuations, while DNS directly resolves all the scales of the eddies without any approximation or parameterization; however, this approach is computationally expensive. LES, which is an intermediate approach between RANS and DNS, is a more common choice, in which the large eddies are explicitly resolved, while the

small eddies are parameterized using Sub-Grid Scale (SGS) schemes (Smagorinsky, 1963; Deardorff, 1970).

Computational Fluid Dynamics (CFD) models resolve turbulence using RANS, LES, or DNS to solve engineering problems involving fluid flows. These models are being increasingly often applied to address urban pollution problems, which has become possible with the availability of large computational resources. The advantage of the CFD approach is that it can resolve the urban morphology (e.g., terrain, buildings, streets, and so on), thus explicitly calculate the eddies cross the streets

and around the buildings. Such models represent the effects of the urban canopy on the dynamics of flows. They also simulate the pollutant retention at street level, usually at a resolution of meters; however, their domain is usually limited to a few



kilometers squared (Tominaga and Stathopoulos, 2013; Zhong et al., 2016). Many studies coupled mesoscale models with CFDs to investigate micro-scale processes in realistic applications (Baik et al., 2009; Zheng et al., 2015). However, due to the large gap between scales in mesoscale and CFD formulations, the processes covering intermediate scales are missing. In

addition, CFD modules mainly focus on dynamical processes in the flow field, but do not describe other major atmospheric processes related to the physics, the thermodynamics, radiative transfer, cloud processes, land surface exchanges, etc.

LES formulations on the other hand can also be applied in meteorological models with full physical and chemical processes in a spatial domain that is considerably larger than a common CFD domain. An illustrative example (Khan et al., 2021) of an LES module with atmospheric chemistry linked to regional information is provided by the PALM-4U (Parallelized Large-

eddy simulation Model for Urban applications) model applied to an urban environment. To perform realistic studies, the PALM model uses a stand-alone pre-processor (INIFOR) that creates initial and boundary meteorological conditions from mesoscale models (Maronga et al., 2020). Recently, efforts by the atmospheric LES community to model urban effects have focused on exploiting the accelerated computing from GPUs (Muñoz-Esparza et al., 2020, 2021; Sauer and Muñoz-Esparza, 2020), since these fine-scale simulations are otherwise computationally too expensive.

The Weather Research and Forecasting (WRF; https://www2.mmm.ucar.edu/wrf/users/; Skamarock, 2019) model allows multiscale nested simulations of the atmosphere with a full suite of atmospheric dynamics, surface exchanges, radiation, and cloud physics. A LES module was embedded into WRF (WRF-LES), which can be applied in two modes: (1) without external forcing from the mesoscale model or by real-world meteorological data (Moeng et al., 2007; Yamaguchi and Feingold, 2012); or (2) by being coupled with the mesoscale WRF (Nozawa and Tamura, 2012; Muñoz-Esparza et al., 2017). The WRF model

can also account for atmospheric chemistry (WRF-Chem; https://ruc.noaa.gov/wrf/wrf-chem/; Grell et al., 2005) and specifically simulate the transport and chemical reactions between various species with different mechanisms. In this work, we use the WRF model framework with the LES mode and online chemistry (WRF-LES-Chem) to perform muti-scale simulations of atmospheric chemical species. The advantage of using WRF-LES-Chem is that all the physical and chemical processes are consistent among the different scales. This allows to analyze the differences induced by turbulent interactions.

Previous studies using coupled WRF-LES-Chem investigated passive tracer dispersion in urban environments (Nottrott et al., 2014; Gaudet et al., 2017). Here we use a more comprehensive chemical mechanism in the coupled WRF-LES-Chem model to study the ozone photochemistry in a complex terrain.

The model experiments were conducted for the city of Hong Kong, a highly urbanized area with severe air pollution. The region is characterized by heterogeneous land use with forest over the mountains and dense constructions along the coast. The

pollution originates from different sources including road transport, navigation, civil aviation, power plants, industry, hill fires, and other combustion and non-combustion emissions. The complex topography, land surface, and diverse emissions make it difficult to predict the air quality patterns and variability in the city, especially in the dense built-up area. Hong Kong therefore poses challenging questions that can be investigated with a high-resolution model and is a perfect scenario to evaluate the capability of WRF-LES-Chem to perform high resolution air pollution predictions.



The purpose of this work is to exploit the WRF framework with embedded LES and chemistry modules to conduct realistic high-resolution air quality simulations in a densely populated urban area. The model setup is presented in Section 2. The description of the observations can be found in Section 3. In Section 4, the performance of the WRF-LES-Chem model is evaluated by comparison to available observations. The influence of spatial resolution on the model results is also presented, including the impact of the resolved turbulence on atmospheric chemistry by comparing it to mesoscale model output with

PBL parameterization. Section 5 summarizes the study and provides conclusions.

## 2 Model setup

### 2.1 Dynamic settings

The WRF model coupled with chemistry (version 4.0.2) is used in the present study. The coupled mesoscale and LES module within WRF are adopted to represent spatial and temporal scales specifically capturing synoptic to turbulent motions. The

model setup consists of seven one-way nested domains, out of which D01-D04 are mesoscale domains and D05-D07 are LES domains (Fig. 1). The horizontal resolutions for the mesoscale domains are 24.3 km, 8.1 km, 2.7 km, and 900 m, respectively from the outer to inner domains, with the number of points in the zonal and meridional directions of $141 \times 141$, $141 \times 141$, $141 \times 141$, and $180 \times 180$ in the corresponding domains. The grid spacings in the LES domains are 300 m, 100 m, and 33.3 m, respectively for D05 to D07, and the corresponding grid numbers are $240 \times 240$, $396 \times 342$, and $243 \times 243$. The vertical

layers are set to 63 levels for all the domains, with the finest vertical resolution of 12.5 m to 50 m in the lowest 1.3 km (30 layers). For the heights above 1.3 km, the vertical spacing increases gradually towards the model top located at 50 hPa.

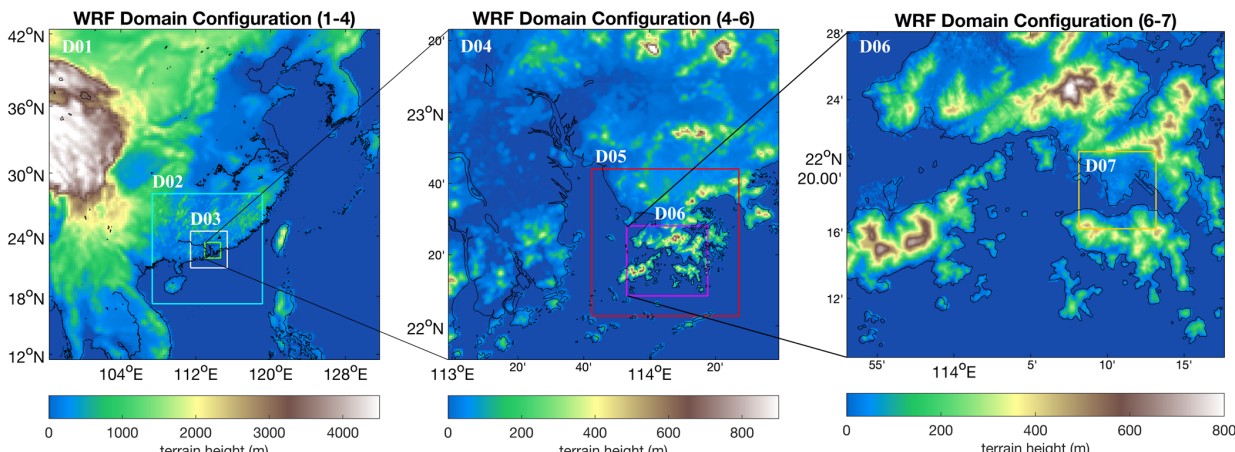

**Figure 1.** Setup of the seven one-way nested WRF domains and a representation of the topography adopted in the model.

The meteorological initial and boundary conditions are obtained from the Final Operational Global Analysis (FNL) data

produced by the National Centers for Environmental Prediction (NCEP) on $1° \times 1°$ grids and 6-hour intervals. The topography data for D01-D05 are from the GMTED2010 (Global Multi-resolution Terrain Elevation Data) 30-arc-second dataset





developed by the U.S. Geological Survey (USGS) and the National Geospatial-Intelligence Agency (NGA). For the two innermost domains D06 and D07, the terrain heights are adopted from the ALOS world 3D data (Takaku et al., 2014) distributed by OpenTopography (https://opentopography.org; last access: Jun 1, 2020) with a spatial resolution of 30 m. The topography maps after pre-processing are shown for different domains in Fig. 1. The land use data are obtained from the Noah-modified 21-category IGBP (International Geosphere-Biosphere Programme) -MODIS (Moderate Resolution Imaging Spectroradiometer) database on 30-arc-second resolution (Friedl et al., 2010).

The Yonsei University (YSU; Hong et al., 2006) PBL scheme is adopted in the mesoscale domains to parameterize the SGS turbulent fluxes within the PBL. The PBL parameterization, however, is turned off in the LES domains, where the large eddies are directly resolved and the effects of the small eddies are expressed by the three-dimensional 1.5-order TKE subgrid closure of Deardorff (Deardorff, 1970; Moeng, 1984). Most of the other main physical parameterizations are identical for both mesoscale and LES domains. They include the RRTM longwave radiation scheme (Mlawer et al., 1997), the Goddard shortwave radiation scheme (Chou, 1994), the Lin microphysics scheme (Lin et al., 1983), the Noah land surface model (Chen and Dudhia, 2001), and the revised MM5 Monin-Obukhov surface layer scheme (Jiménez et al., 2012). For the coarser mesoscale simulations (D01-D03), the Grell-Freitas cumulus scheme (Grell and Freitas, 2014) is used, while it is turned off in the innermost mesoscale domain (D04) and the LES domains.

The simulations of the mesoscale WRF and WRF-LES are performed separately. The mesoscale simulations run from 00:00 UTC on July 30, 2018, to 12:00 UTC on August 1, 2018, and the first 45 hours are considered as a spin-up period. This longtime spin-up ensures that the atmosphere is in balance with the new sea surface temperature and soil properties, and that the atmospheric chemistry reaches an equilibrium state. The output from the innermost mesoscale domain D04 with a time interval of 10 min is taken as the initial and boundary condition for the LES runs. The LES simulations run from 21:00 UTC on July 31, 2018, to 12:00 UTC on August 1, 2018, which corresponds to the Hong Kong local time (LT) of 05:00 to 20:00 on August 1, 2018. To accelerate the generation of the turbulence in the outer LES domain, the cell perturbation method (Muñoz-Esparza et al., 2014, 2015, 2017) is applied to D05. Since the grid space of D05 is 300 m, which is too coarse for LES, we consider this case to represent an intermediate domain to pass synoptic motions from mesoscale domains (with turbulence generated in D05) to the smaller LES domains. Therefore, the analysis will focus on the innermost mesoscale domain D04 and the fine LES domains D06 and D07. The first hour from the LES simulations is considered as spin-up time. The output intervals for LES domains are 5 min for D05, 2 min for D06, and 1 min for D07.

## 2.2 Chemical settings

In the present study, we adopt the RADM2 (Regional Acid Deposition Model, 2nd generation; Stockwell, 1990) chemical mechanism with 63 chemical species and 157 reactions. While most chemical species are transported in the model, several fast-reactive species such as the organic peroxy radicals are assumed to follow local photochemical equilibrium conditions. When considering the turbulent flow in the LES simulations with an integration time shorter than the chemical reaction time, the fluctuations in the concentration of these species in the flow, which result in the segregation effect, cannot be ignored.





Therefore, the transport of the fast-reactive radicals is turned on in the LES domains. The TUV radiation transfer scheme is used for the calculation of the photolysis rates. The initial and boundary condition for the chemical species is taken from the CAMChem (Community Atmosphere Model with chemistry; Emmons et al., 2020) output (Buchholz et al., 2019).

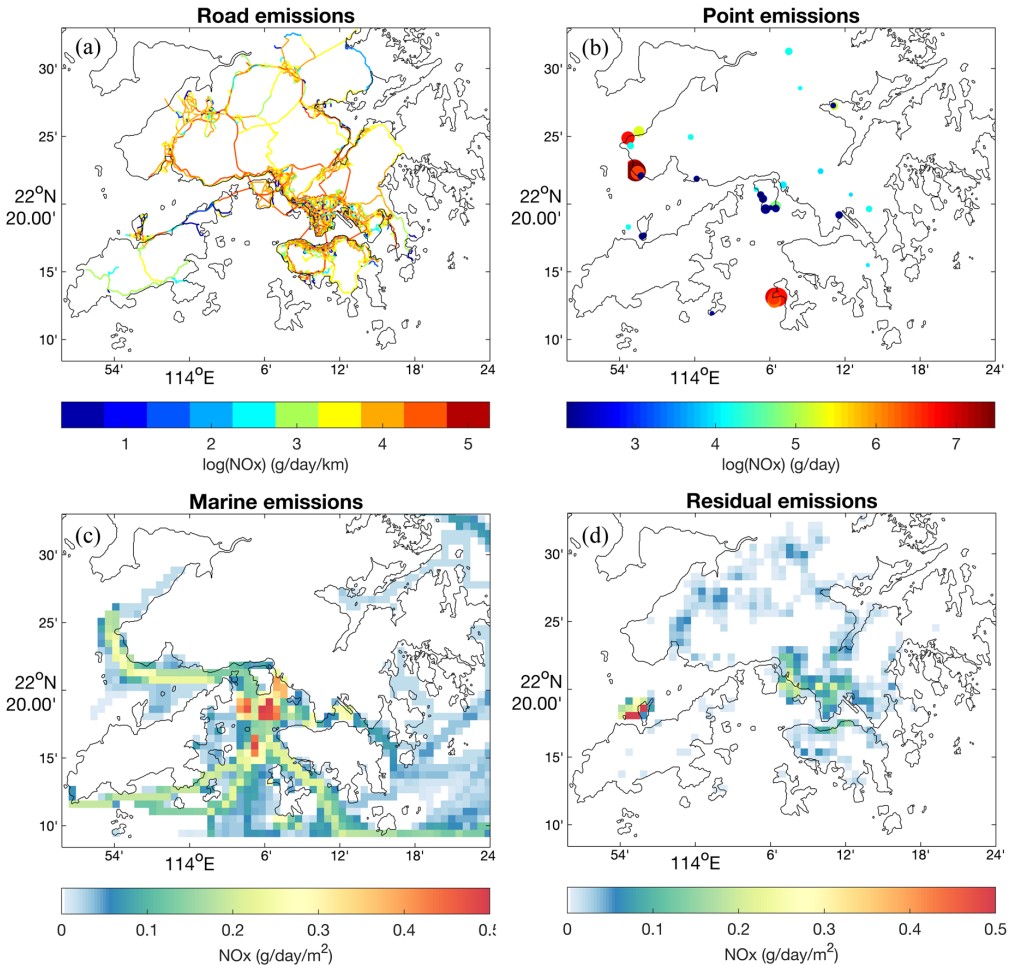

**Figure 2.** Emission maps of $NO_X$ provided by HKEPD for the Hong Kong region from road (a), point (b), marine (c), and residual (d)
sources. The road emissions are shown as line sources for every road with the unit of g day$^{-1}$ km$^{-1}$. The point sources include power plants, industries, crematorium, and tank farms, with the unit of g day$^{-1}$ per point. The sizes of the points indicate the height of the source. The marine and residual emissions are area sources provided with a 1 km × 1 km resolution with the unit of g day$^{-1}$ m$^{-2}$.

To cover the entire simulation domains, the anthropogenic emissions adopted in our simulations include the Multi-resolution Emission Inventory for China (MEIC; http://meicmodel.org, last access: 13 May 2022; Zheng et al., 2018) for the year 2017,
the MIX emissions for Asia outside of China for the year 2010 (Li et al., 2017), and the international ship emissions from the Emissions Database for Global Atmospheric Research (EDGAR; https://edgar.jrc.ec.europa.eu, last access: 13 May 2022; Johansson et al., 2017) based on the year 2015. We use these emission inventories for the specific year because it is the most updated year that is publicly available. In order to have more detailed emission information in the focused area, we also employ



the Pearl River Delta region (PRD) emissions with a resolution of 3 km based on the year 2014 (Zheng et al., 2009; Bian et
al., 2019). The emission data for Hong Kong is provided by the Hong Kong Environmental Protection Department (EPD) for
scientific research purposes, which includes road-based vehicle emissions for the year 2014, point emissions of power plants,
industries, crematorium, and tank farms for the year 2017, marine emissions with a resolution of 1 km for the year 2015, and
residual emissions (total emissions subtract the road, marine, and point emissions) on 1 km resolution for the year 2015. The
emission maps of NOx from different sources are shown in Fig. 2 as an illustrative example. The Hong Kong emissions are
scaled       to      the       simulation       year      of      2018       according       to       the       annual      trend
(https://www.epd.gov.hk/epd/english/environmentinhk/air/data/emission_inve.html, last access 13 May 2022). The biogenic
emissions are calculated with the online Megan model (Model of Emissions of Gases and Aerosols from Nature; Guenther et
al., 2006) embedded in WRF-Chem.

## 3 Observational data

### 3.1 Surface monitoring network

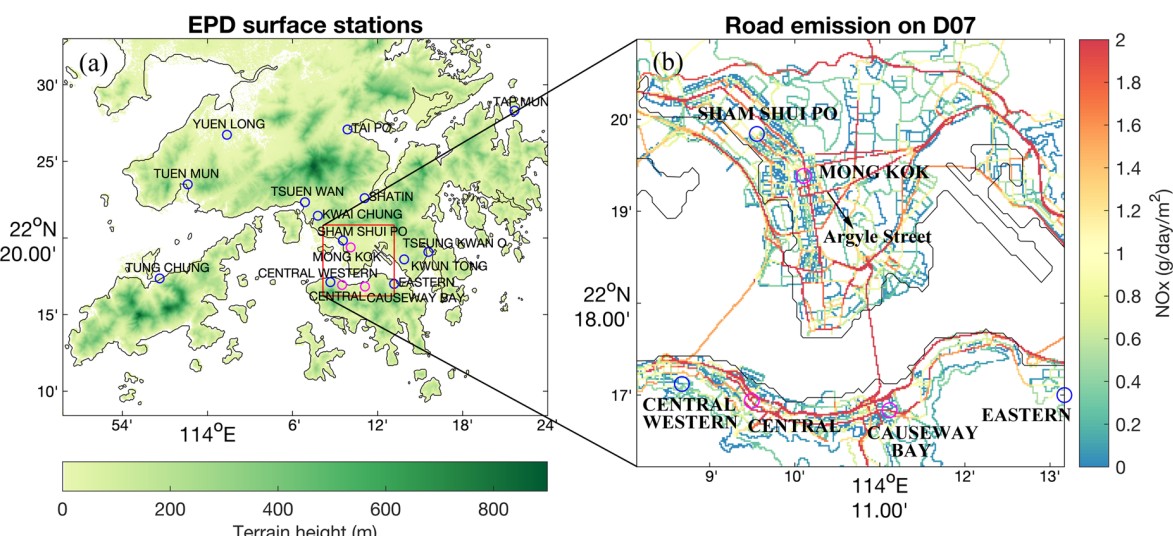

**Figure 3.** Locations of the surface monitoring stations represented on the map of the Hong Kong topography (a) and the road emissions
interpolated to D07 domain grids (b). The blue circles represent the general stations and the magenta circles the roadside stations.

Hourly-averaged     air     quality     monitoring     data     are     obtained     from     the     Hong     Kong     EPD's     network
(https://cd.epic.epd.gov.hk/EPICDI/air/station/?lang=en, last access: 13 May 2022). Among all the stations, there are three
roadside stations (Mong Kok, Central, and Causeway Bay) and 13 general stations (relative to roadside stations; see Fig. 3).
The information about the stations is listed in Table 1. The mesoscale domains (D01-D04) and the outer LES domain (D05)
cover all the stations. For the smaller LES domains, D06 covers all the stations except of Tap Mun station, and D07 only
covers the roadside stations and three general stations. The instrument heighs are below 5 m above ground for the roadside



stations, while they are between 10 m and 30 m for the general stations. We use the first layer of the model data to compare between the surface measurements and the model simulations. During the simulation period, no $NO_X$ measurements are available at the Eastern station, and some data are also missing around noon time at Tsuen Wan and Kwai Chung stations.

**Table 1.** List of surface stations in Hong Kong EPD's network

| Station | Type | Latitude [° N] | Longitude [° E] | Altitude above ground [m] | Covered by domains |
|---------|------|----------------|-----------------|---------------------------|--------------------|
| TAP MUN | General | 22.4713 | 114.3607 | 11 | 1-5 |
| YUEN LONG | General | 22.4452 | 114.0227 | 25 | 1-6 |
| TAI PO | General | 22.4510 | 114.1646 | 28 | 1-6 |
| TUEN MUN | General | 22.3912 | 113.9767 | 27 | 1-6 |
| TUNG CHUNG | General | 22.2889 | 113.9437 | 27.5 | 1-6 |
| KWAI CHUNG | General | 22.3571 | 114.1296 | 13 | 1-6 |
| TSUEN WAN | General | 22.3718 | 114.1145 | 17 | 1-6 |
| SHATIN | General | 22.3763 | 114.1845 | 25 | 1-6 |
| KWUN TONG | General | 22.3096 | 114.2312 | 14.7 | 1-6 |
| TSEUNG KWAN O | General | 22.3177 | 114.2596 | 16 | 1-6 |
| SHAM SHUI PO | General | 22.3302 | 114.1591 | 17 | 1-7 |
| CENTRAL/WESTERN | General | 22.2849 | 114.1444 | 16 | 1-7 |
| EASTERN | General | 22.2829 | 114.2194 | 15 | 1-7 |
| MONG KOK | Roadside | 22.3226 | 114.1683 | 3 | 1-7 |
| CENTRAL | Roadside | 22.2818 | 114.1581 | 4.5 | 1-7 |
| CAUSEWAY BAY | Roadside | 22.2801 | 114.1851 | 3 | 1-7 |


### 3.2 Ozone sounding profile

The ozone sounding measurements at King's Park station (22.311° N, 114.172° E) in Kowloon peninsula were conducted by Hong Kong Observatory (HKO). We downloaded the measurement data used for model comparison in this study from the World Ozone and Ultraviolet Radiation Data Centre (WOUDC; https://woudc.org/home.php, last access: 13 May 2022). The

profile for the simulation period was derived at 05:55 UTC on 1 August 2018, which corresponds to noon time (13:55 LT), when the convective boundary layer is well developed. The trajectory of the sounding is shown in Fig. 4. For the comparison between models and sounding measurements, the vertical profiles are usually taken at the location of the site, because of the coarse resolution of the models. In fact, the sounding balloon drifted over large horizontal distances in response to the background winds during its ascent. In the case adopted in the present study, the balloon drifted about 1.4 km away from its

release location before it reached the level of 2 km, and it drifted over a distance of 8.6 km at 10 km altitude. Therefore, as for the high-resolution model simulations in this study, we adopt the nearest model grids along the sounding trajectory for a more precise comparison.





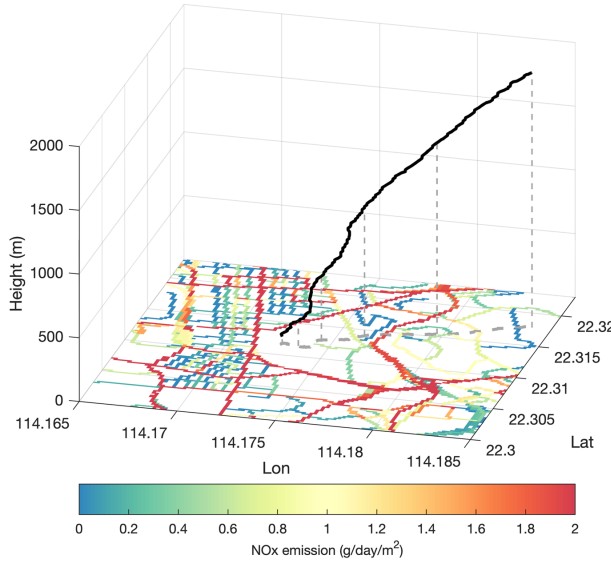

**Figure 4.** Ozone sounding trajectory in the first 2 km over the road emission map adopted for the innermost domain D07.

## 4 Results

### 4.1 General PBL structure

We first evaluate the development of the boundary layer structure in the simulations by comparing model results with sounding profiles at 13:55 LT (Fig. 5). A sharp inversion can be seen in the profiles of the potential temperature and water vapor near the top of the boundary layer, of which this height is usually used to quantify the development of the PBL (Garratt, 1994). For the mesoscale simulations, the PBL height (PBLH) in the model is diagnosed by the YSU PBL scheme. The YSU PBLH for the mesoscale domain (D04, Δ = 900 m) at the nearest grid of the sounding location is 992 m above the surface as shown by the dotted horizontal bar on the D04 profile. In the case of the sounding profiles and the LES model results, the PBLH is determined by applying the bulk Richardson number ($Ri_b$) method (Hong et al., 2006), which is similar to approach used in the YSU scheme. The bulk Richardson number is calculated by

$$Ri_b = \frac{g[\theta_v(z) - \theta_{vs}]z}{\theta_{vs}[u(z)^2 + v(z)^2]} \tag{1}$$

where $\theta_v$ is virtual potential temperature, $u$ and $v$ are horizontal wind components, $g$ is gravitational acceleration; the subscript "s" refers to conditions at the surface. The PBLH is defined as the height where $Ri_b$ first exceeds a threshold value of zero (Hong et al., 2006). The PBLH derived from the relation (1) for D04 is 970 m, which is very similar to the value (992 m) simulated by the YSU scheme. The calculated PBLH are 1132 m and 1204 m for the D06 (Δ = 100 m) and the D07 (Δ = 33.3 m) LES domains, respectively, implying that high-resolution enhances the turbulent mixing and hence raises the boundary layer height. The PBLH derived from the observation profile is 1141 m, which is close to the LES simulations but higher than the value derived in the mesoscale domain D04.





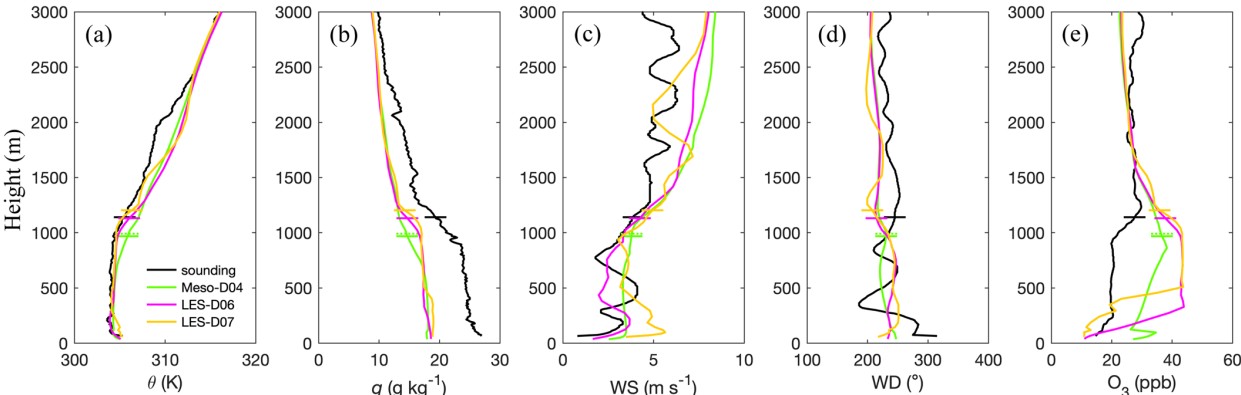

**Figure 5.** Comparison between ozone sounding measurements and model simulations at 13:55 LT. The variables are the potential
temperature ($\theta$; a), the water vapor mixing ratio ($q$; b), wind speed (WS; c), wind direction (WD; d), and ozone mixing ratio ($O_3$; e). The
black lines represent observations; the green lines the simulations from D04 with resolution of 900 m; the magenta lines the simulations
from D06 with resolution of 100 m; and the yellow lines the simulations from D07 with resolution of 33.3 m. The solid horizontal bars
indicate the calculated PBLH for each domain, and the dotted horizontal bars show the simulated PBLH from YSU scheme.

Potential temperature is generally satisfactorily represented by the simulations for both the mesoscale and the LES domains.
However, the mesoscale simulation overestimates potential temperature near the PBLH, which explains why the calculated
PBLH is lower than the observed value. This suggests that the convection in the mesoscale model is relatively weak at this
location. All of the simulations underpredict water vapor consistently among all simulation domains, especially at low
altitudes; however, this is consistent among all simulation domains. The basic structure of the wind profiles is reasonably well
captured by the simulations, but the large fluctuations caused by local turbulence are difficult to replicate by the model in a
deterministic manner. Both mesoscale and LES-scale simulations are able to capture the observed southwest wind directions
in the lower troposphere, except the near surface northwest winds. This deviation between simulated and observed winds in
the near surface layer is likely due to the local urban morphology effects, which are not included in the WRF model. Overall,
the coupled mesoscale to LES-scale WRF model is found to be capable of reproducing realistic atmospheric conditions for the
period of interest.

## 4.2 Comparison of ozone profile with sounding measurements

The simulated vertical profiles of ozone are compared to the ozone sounding measurements (Fig. 5). The mesoscale simulation
overestimates the ozone concentrations near the surface, probably due to the lack of capability to resolve the road emissions.
The emitted NO from traffic sources reacts with $O_3$ to produce $NO_2$:

$$NO + O_3 \rightarrow NO_2 + O_2 \tag{R1}$$

Under high pollution conditions, this leads to an increase in $NO_2$ concentrations and a decrease in the $O_3$ level. Therefore, the
small emission of NO in the coarse resolution model results in a weak titration of ozone and hence an overestimation of the
ozone concentration. The LES simulations tend to give a lower level of surface $O_3$ due to the better representation of the spatial
emission distribution and improves the agreement with the measurements, illustrating the advantage of the high-resolution



LES. Both mesoscale and LES simulations overestimate the $O_3$ concentrations within the rest of the PBL above the surface

layer, which implies that the $O_3$ source is overestimated in this region. The formation of $O_3$ follows the photolysis of $NO_2$:

$NO_2 + hv \rightarrow NO + O(^3P)$                      (R2)

$O(^3P) + O_2 + M \rightarrow O_3 + M$                   (R3)

The combination of reactions (R1) – (R3) results in a null cycle without any net gain/loss of $O_3$. However, in the presence of

the volatile organic compounds (VOCs), the $RO_X$ ($RO_X$ = OH + $HO_2$ + $RO_2$; $RO_2$ stands for any organic peroxy radical)

photochemical cycle continuously supplies $HO_2$ and $RO_2$ to oxidize NO into $NO_2$ without consuming ozone. This mechanism

limits the production of $O_3$ in the troposphere (Wang et al., 2017):

$OH + RH + O_2 \rightarrow RO_2 + H_2O$                  (R4)

$RO_2 + NO \rightarrow RO + NO_2$                    (R5)

$RO + O_2 \rightarrow HO_2 + carbonyls$                  (R6)

$HO_2 + NO \rightarrow OH + NO_2$                    (R7)

where RH represents any non-methane hydrocarbon.

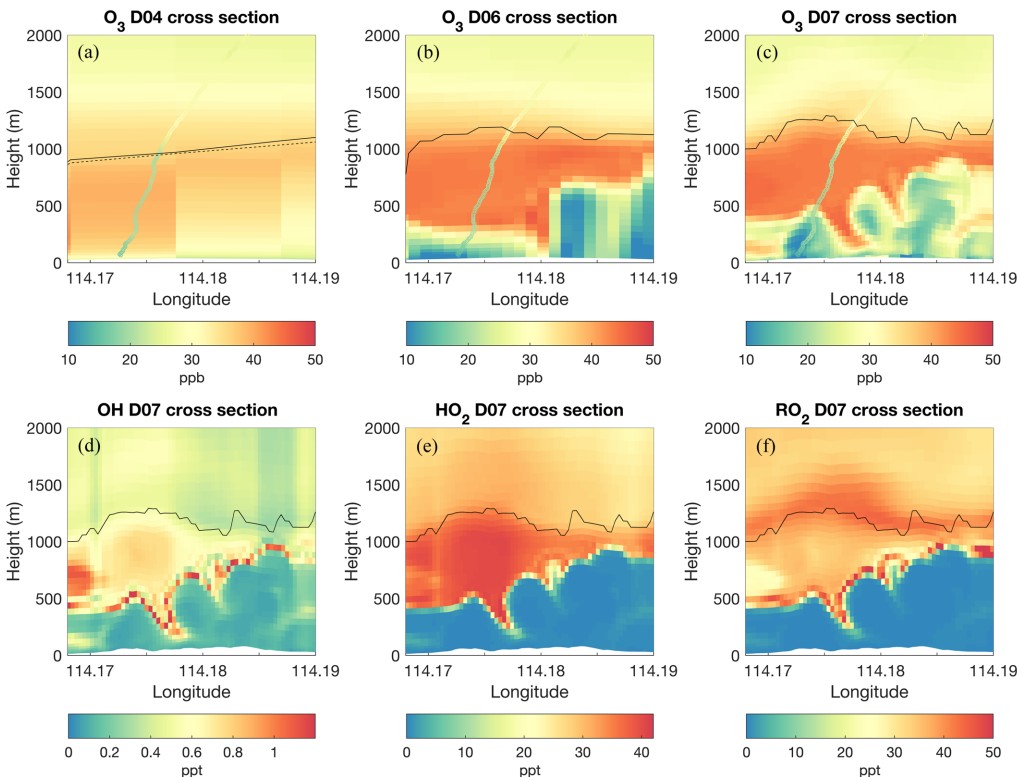

**Figure 6.** Top: $O_3$ mixing ratio (ppbv) cross sections along the balloon sounding trajectory for the D04 (a), D06 (b), and D07 (c) domains. Thick colored lines represent the ozone sounding measurements along the trajectories at 13:55 LT. The black solid lines are PBLH calculated

using the bulk Richardson number method, and the black dash line is the PBLH from the model output for the mesoscale domain, which is simulated in the YSU PBL scheme. Bottom: Cross sections along sounding trajectory for D07 of the OH (d), $HO_2$ (e), and $RO_2$ (f; sum of the organic peroxy radicals in RADM2 chemical mechanism) mixing rations (pptv).





To identify the possible reasons for the disagreement between simulations and observations, the vertical cross sections of the $O_3$ mixing ratios along the sounding trajectory for the three domains (D04, D06, and D07) are shown in Fig. 6. This graph shows that the $O_3$ concentrations derived by the mesoscale model are too high within the entire boundary layer depth. In the coupled LES simulations, lower $O_3$ concentrations can be seen near the surface, which agrees with the measurements. The low $O_3$ air masses from the surface can reach approximately 80% of the PBL height, indicating a strong vertical mixing. In addition, the air parcels with low/high $O_3$ values are clearly transported to higher/lower altitudes by the updrafts/downdrafts produced by the stronger turbulent mixing occurring in the high-resolution LES domain D07 (see Fig. 6c). In addition to the low $O_3$ near the surface, high $O_3$ air masses can be found in the west of the domain, resulting in an overestimation of $O_3$ between 400 m and 1000 m. The cross sections of OH, $HO_2$, and $RO_2$ are also depicted in Fig. 6 (bottom row), which shows high concentrations of OH, $HO_2$, and $RO_2$ along with the high $O_3$ concentrations. As discussed above, the $RO_X$ cycle (R4) – (R7) converts NO to $NO_2$, and tends to produce $O_3$ under such conditions. This results in high $O_3$ in the western part of the domain and in the upper boundary layer. The comparison between the models and the observations indicates that the $O_3$ distribution in the LES simulations is offset relative to the measurements – the modeled $O_3$ is shifted to the downwind direction. Fig. 5c shows that the model generally overestimates the horizontal wind speed in the boundary layer, and therefore displaces the air masses with low $O_3$ too rapidly to the east of the sounding trajectory, which probably explains the overestimation of $O_3$ between 400 m and 1000 m.

**4.3 Comparison of pollutants with surface observations**

Figure 7 shows the comparison between simulated $NO_X$ concentrations and surface measurements. We focus here on the model results from the innermost mesoscale domain (D04, 900 m, green dots), the middle LES domain (D06, 100 m, magenta dots), and the inner most LES domain (D07, 33.3 m, yellow dots). At most general stations, the mesoscale simulation of D04 captures well the mean $NO_X$ concentrations and the diurnal variations. The best agreement is found at the stations located in the northern and western suburban areas of Hong Kong, i.e., stations that are far away from the city center including Tap Mun, Yuen Long, Tai Po, Tuen Mun, Tung Chung, and Kwai Chung. The mean $NO_X$ concentrations at these suburban stations are lower than 100 $\mu$g m$^{-3}$. This indicates that the regional models are good at predicting the background levels of the pollutants over a relatively large area. For the other general stations closer to the emission sources, the simulated $NO_X$ concentrations in the D04 case are higher than the measured values except at the Kwun Tong station. The overestimation of $NO_X$ mainly occurs during the rush hours (7:00-9:00 and 18:00-20:00 LT), implying that the road emissions are overestimated at these locations. As for the roadside stations, as expected, the $NO_X$ values are underestimated in the mesoscale simulation due to the poor representation (too coarse resolution) of the road emissions. The mean biases (MB) and root mean square errors (RMSE) of the simulated against the measured $NO_X$ concentrations are listed in Table 2. The smallest MBs are found at the suburban stations, most of which are below 15 $\mu$g m$^{-3}$. The RMSEs at these stations are between 15 $\mu$g m$^{-3}$ and 30 $\mu$g m$^{-3}$. At the other general stations, the only negative MB is found at Kwun Tong station, where the mesoscale D04 simulation underestimates the observed $NO_X$. The MBs at the other general stations are all positive, ranging between 30 $\mu$g m$^{-3}$ and 50 $\mu$g m$^{-3}$. The



RMSEs at the general stations close to the city center are $40 - 60$ $\mu g$ m$^{-3}$. The MBs and RMSEs for the roadside stations are much higher than those of the general stations. The MBs are all negative at the roadside stations with values of -97.006 $\mu g$ m$^{-3}$ at Mong Kok, -54.373 $\mu g$ m$^{-3}$ at Central, and -171.499 $\mu g$ m$^{-3}$ at Causeway Bay stations, respectively. The RMSEs range from 117.431 $\mu g$ m$^{-3}$ to 219.693 $\mu g$ m$^{-3}$ for the roadside stations.

The 100-m LES (D06) simulation results in mean concentrations and diurnal variations of NO$_X$ that are similar to those from the mesoscale model at the suburban stations, but with larger variances resulting from the explicitly resolved eddies. Some improvements can be seen during evening hours at Tai Po and Tung Chung. The calculated MBs and RMSEs show improvement in the D06 LES simulation at some suburban stations when compared to the mesoscale simulation. For example, at Tai Po station, the MB decreases from 5.574 $\mu g$ m$^{-3}$ in the D04 simulation to 1.196 $\mu g$ m$^{-3}$ in the D06 simulation, and the

RMSE decreases from 21.846 $\mu g$ m$^{-3}$ to 13.667 $\mu g$ m$^{-3}$. The variances of the simulated NO$_X$ concentrations from D06 are larger at the stations closer to the city center, indicating the complexity of the mixing and dispersion of pollutants in the turbulent flows over heterogenous urban surfaces even without the explicit representation of buildings. The agreement between the NO$_X$ concentrations simulated by the 100-m LES and by the measurements is improved at some urban general stations with smaller MBs and RMSEs (e.g., Shatin and Central Western), and is worsened at some other general stations (e.g., Tsuen

Wan, Kwai Chung, Kwun Tong, and Sham Shui Po). In other words, the high-resolution model does not necessarily provide much better predictions of the chemical species at the general stations. One possible explanation is that the emission inventories (or their spatial and temporal resolution) are not sufficiently accurate and resolved in time. For instance, the road emissions are based on annual mean traffic flows, which do not capture the real-time variations. Also, while higher resolution than standard mesoscale predictions, the LES domains used here do not allow explicit resolution of buildings, which is expected to

have a considerable impact in capturing local effects within the urban canopy.  As for the roadside stations, the improvements of the 100-m LES simulation are clearer. The D06 simulation provides larger NO$_X$ values than the D04 simulation and agrees more closely with the measurements. This can be seen from the decreased negative biases: MB decreases by about 60 $\mu g$ m$^{-3}$, 50 $\mu g$ m$^{-3}$, and 30 $\mu g$ m$^{-3}$, at Mong Kok, Central, and Causeway Bay, respectively. The RMSEs are also reduced at the three roadside stations. However, there are still disagreements between the D06 simulations and the measurements, for example, the

case at Causeway Bay and, between 9:00 – 12:00 LT, at Central station.







**Figure 7.** Comparison between $NO_X$ surface measurements and model simulations. Black circles are hourly observations; small dots with different colors (green: D04; magenta: D06; magenta: D07) represent the simulations at output intervals. Dots with error bars refer to the hourly averages with standard deviations. Note that the y-axis is different for the general and roadside stations.





**Figure 8.** Same as Fig. 7 but for ozone concentrations.





**Table 2.** Statistical parameters to evaluate the modeled NO$_X$ against surface measurements in Hong Kong EPD's network. Both mean bias
(MB) and root mean square error (RMSE) are with the unit of $\mu$g m$^{-3}$.

| Station | MB ($\mu$g m$^{-3}$) | | | RMSE ($\mu$g m$^{-3}$) | | |
|---|---|---|---|---|---|---|
| | D04 | D06 | D07 | D04 | D06 | D07 |
| General | | | | | | |
| TAP MUN | 24.368 | - | - | 27.492 | - | - |
| YUEN LONG | 8.271 | 2.538 | - | 15.454 | 13.744 | - |
| TAI PO | 5.574 | 1.196 | - | 21.846 | 13.667 | - |
| TUEN MUN | 1.118 | 17.397 | - | 27.455 | 24.888 | - |
| TUNG CHUNG | 14.370 | 3.712 | - | 18.724 | 24.902 | - |
| KWAI CHUNG | 4.913 | 76.343 | - | 29.672 | 85.620 | - |
| TSUEN WAN | 46.593 | 72.477 | - | 55.042 | 76.611 | - |
| SHATIN | 34.596 | 30.122 | - | 43.717 | 39.145 | - |
| KWUN TONG | -19.761 | 46.779 | - | 55.451 | 73.987 | - |
| TSEUNG KWAN O | 31.533 | 37.823 | - | 44.669 | 52.764 | - |
| SHAM SHUI PO | 45.230 | 70.713 | 109.138 | 56.432 | 76.080 | 117.104 |
| CENTRAL/WESTERN | 36.116 | 23.551 | 6.627 | 55.847 | 36.254 | 27.552 |
| EASTERN | - | - | - | - | - | - |
| Roadside | | | | | | |
| MONG KOK | -97.006 | -38.703 | -20.663 | 117.431 | 70.419 | 52.698 |
| CENTRAL | -54.373 | -4.965 | 52.005 | 126.333 | 88.712 | 85.511 |
| CAUSEWAY BAY | -171.499 | -139.228 | -92.759 | 219.693 | 194.115 | 171.747 |

When the LES grid size is decreased to 33.3 m (D07), the agreement between the simulation and the observations is further improved at the roadside stations. Large improvements can be seen at Central station in the morning; however, the agreement is worse in the evening. This results in a larger mean bias at the Central station compared to the D06 simulation. The MB at
Mong Kok decreases from -38.703 $\mu$g m$^{-3}$ in the D06 simulation to -20.663 $\mu$g m$^{-3}$ in the D07 simulation, and it reduces from -139.228 $\mu$g m$^{-3}$ to -92.759 $\mu$g m$^{-3}$ at Causeway Bay station. For the two general stations covered by the D07 domain (note that Eastern station is covered by D07 but has no NO$_X$ data for the simulation day), the D07 simulation improves the agreement with the measurements at Central Western, with the corresponding MB reduces from 23.551 $\mu$g m$^{-3}$ to 6.627 $\mu$g m$^{-3}$ and the RMSE decreases from 36.254 $\mu$g m$^{-3}$ to 27.552 $\mu$g m$^{-3}$. However, Both MB and RMSE increase at Sham Shui Po station. The
comparison between simulations and observations indicates that increasing the model resolution does not necessarily lead to an improvement over the background air quality simulation, although it can provide a better representation of the air pollution structure over complex land surfaces and near the emission sources.

Next, we compare the simulated ozone concentrations against the observations (refer to Fig. 8), and the statistical skill metrics are listed in Table 3. The interpretation of ozone simulations is more complex because, as opposed to the case of a
primary pollutant with a short lifetime like NO$_X$, which is largely controlled by surface emissions, O$_3$ is also affected by chemical transformations, dynamic transport, and turbulent mixing. As seen from most of the stations in Fig. 8, the diurnal evolution of the surface O$_3$ is characterized by low values in the morning and evening, and by high values during daytime with a concentration peak occurring around 14:00 LT, when the photochemical production is largest. The lower O$_3$ concentrations



at the roadside stations compared to those at the general stations is of characteristic of an environment that is $NO_X$ saturated
and hence VOC limited. Overall, the mesoscale simulation from D04 matches the observations well at the general stations,
suggesting that the representation of the physical and chemical processes in the model is sufficiently realistic. The D04
simulation slightly overestimates the $O_3$ concentrations at most general stations during daytime, but underestimates $O_3$ in the
morning and evening, which is partly attributed to the $NO_X$ overestimation during the rush hour, resulting in the overestimated
titration of ozone by NO. The positive MBs at the general stations mostly range from 0.6 $\mu g$ m$^{-3}$ to 6.6 $\mu g$ m$^{-3}$ except at Tung
Chung station near the airport in the west of Hong Kong where it reaches 12.970 $\mu g$ m$^{-3}$. The negative MBs are in the range
between -0.331 $\mu g$ m$^{-3}$ and -4.463 $\mu g$ m$^{-3}$, and mostly occur at the stations in the east of Hong Kong, implying that the model
underestimates the $O_3$ concentration in that region, possibly due to the strong transport of $NO_X$ towards the east by the westerly
winds as discussed in Section 4.2. The RMSEs at the general stations are estimated to be between 8.416 $\mu g$ m$^{-3}$ and 25.257 $\mu g$
m$^{-3}$, and the largest value is found at the Eastern station. The $O_3$ concentrations are overestimated by the mesoscale simulation
at the roadside stations as a consequence of the low $NO_X$ traffic emissions in the coarse model. The MBs are 11.671 $\mu g$ m$^{-3}$ at
Mong Kok, 6.208 $\mu g$ m$^{-3}$ at Central, and 9.830 $\mu g$ m$^{-3}$ at Causeway Bay, and the corresponding RMSEs are 18.079 $\mu g$ m$^{-3}$,
13.792 $\mu g$ m$^{-3}$, and 18.551 $\mu g$ m$^{-3}$, respectively.

**Table 3.** Same with Table 2, but for $O_3$

| Station | MB ($\mu g$ m$^{-3}$) | | | RMSE ($\mu g$ m$^{-3}$) | | |
|---|---|---|---|---|---|---|
| | D04 | D06 | D07 | D04 | D06 | D07 |
| General | | | | | | |
| TAP MUN | -0.331 | - | - | 14.271 | - | - |
| YUEN LONG | 4.054 | 7.589 | - | 8.416 | 12.304 | - |
| TAI PO | 5.110 | 3.777 | - | 19.036 | 13.099 | - |
| TUEN MUN | 6.584 | -1.392 | - | 11.322 | 10.105 | - |
| TUNG CHUNG | 12.970 | 15.428 | - | 19.392 | 19.368 | - |
| KWAI CHUNG | 1.663 | -6.121 | - | 8.556 | 9.410 | - |
| TSUEN WAN | 0.616 | -8.730 | - | 12.635 | 12.517 | - |
| SHATIN | -0.832 | -3.868 | - | 13.134 | 15.578 | - |
| KWUN TONG | 4.562 | -3.340 | - | 14.191 | 8.822 | - |
| TSEUNG KWAN O | -1.323 | -7.101 | - | 14.319 | 12.853 | - |
| SHAM SHUI PO | 2.238 | -10.183 | -13.115 | 12.184 | 12.450 | 16.375 |
| CENTRAL/WESTERN | -4.463 | -1.763 | 11.644 | 19.555 | 13.232 | 17.681 |
| EASTERN | -3.195 | -5.646 | -4.958 | 25.257 | 17.593 | 18.279 |
| Roadside | | | | | | |
| MONG KOK | 11.671 | -0.254 | 1.127 | 18.079 | 4.979 | 6.054 |
| CENTRAL | 6.208 | 1.800 | -0.809 | 13.792 | 6.626 | 9.488 |
| CAUSEWAY BAY | 9.830 | 5.371 | 6.129 | 18.551 | 12.674 | 11.929 |

When the LES simulation is performed at a 100-m resolution (D06), the $O_3$ concentrations are reduced at most of the general
stations except at Yuen Long, Tung Chung, and Central/Western station. Since the mesoscale simulation overestimates the
observed $O_3$ concentrations, this reduction in the $O_3$ concentration slightly improves the agreement with measurements. The





LES also reduces the ozone discrepancies in the evening at Tai Po, Tung Chung, and Central Western stations. However, the calculated MBs and RMSEs do not show consistent improvement. As for the roadside stations, where lower level of $O_3$ concentration is often observed, the simulated $O_3$ concentrations reproduce the observation better especially at Mong Kok. With the D06 simulation, the agreement at Central station is also improved, however, with some discrepancies during the morning rush hour. The improvement at Causeway Bay is relatively limited. Both MBs and RMSEs decreased compared to the D04 simulation. The MB is reduced from 11.671 $\mu$g m$^{-3}$ to -0.254 $\mu$g m$^{-3}$ at Mong Kok, and from 6.208 $\mu$g m$^{-3}$ to 1.800 $\mu$g m$^{-3}$ at Central station; the change ranges from 9.830 $\mu$g m$^{-3}$ to 5.371 $\mu$g m$^{-3}$ at Causeway Bay. The reductions of RMSEs are approximately 13 $\mu$g m$^{-3}$, 7 $\mu$g m$^{-3}$, and 6 $\mu$g m$^{-3}$, for Mong Kok, Central, and Causeway Bay, respectively.

In the case of ozone, the high-resolution LES simulation (D07) does not provide improved results at the three general stations and even slightly degrades the model performance relative to the D06 simulation. The MBs from the D07 simulation at Sham Shui Po and Central/Western are larger than those from D06, and the MB at Eastern station decreases slightly from -5.646 $\mu$g m$^{-3}$ for D06 to -4.958 $\mu$g m$^{-3}$ for D07. The RMSEs increase modestly at all the three general stations. However, as shown by Fig. 8, some improvement can be noticed at the roadside stations. For instance, in the D07 simulation the $O_3$ concentrations are reduced during the morning hours at Central station, showing a better match with the observation. Although the hourly mean values of $O_3$ from D07 are still too high in the afternoon at Causeway Bay, the observed values are in the range of the variability produced by the high-frequency model output. The MB decreases from 1.800 $\mu$g m$^{-3}$ to -0.809 $\mu$g m$^{-3}$ at the Central station, while the MBs slightly increase at Mong Kok and Causeway Bay. The presented comparison with surface measurements highlights the existence of larger fluctuations in the pollutant concentrations, which the high-resolution LES model is able to capture in response to more complex turbulent flows in the urban area. These fluctuations may bring the concentrations of the chemical species closer to the range of measurements, without substantially modifying the average values derived from the statistical analysis of the LES output.

## 4.4 Spatial distribution of the pollutants

Having validated the model with observations, we now analyze the representation of the spatial distribution of pollutants at different model resolutions. The horizontal distributions of NO, $NO_2$, and $O_3$ near the surface for three different model resolutions are shown in Fig. 9 and Fig. 10, at 6:00 LT (morning) and at 13:00 LT (noon), respectively. These two times are selected to represent different boundary layer conditions and traffic emissions. To better characterize the differences between model resolutions, we examine our results only in the subregion of D07, which is located in the city center of Hong Kong and is subject to strong traffic pollution. This region corresponds to the area covered by D07, but with the south and west edges of the domain removed to limit the influence from the mountains and the ocean.





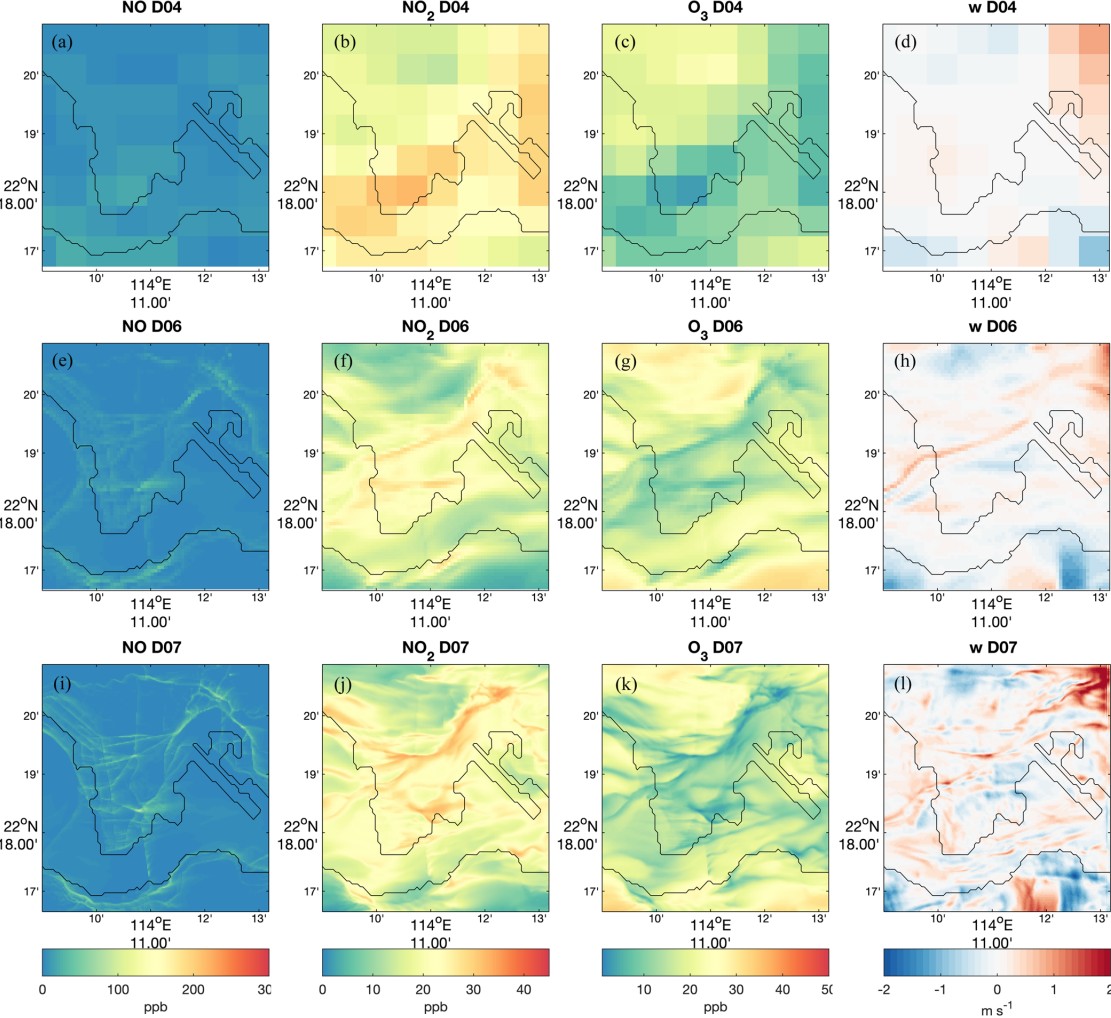

**Figure 9.** Horizontal distribution of simulated NO (first column), NO$_2$ (second column), and O$_3$ (third column) near surface at 6:00 LT for the D04 (first row), the D06 (second row), and the D07 (third row) cases. The fourth column provides the horizontal distribution of vertical wind (w) at approximately 100 m above ground.

At 6:00 LT, around the sunrise, the NO concentrations are very low, because NO has been consumed by the O$_3$ during the night and the photolysis of NO$_2$ is still weak at 6:00 LT. Additionally, the rush hour usually starts at 8:00 LT, so that the direct emission of NO from the road traffic is weak. In the case of the mesoscale domain, the horizontal distribution of chemical species in D04 appears on the figure as mosaic tiles with little spatial variations due to its coarse model resolution (see the first row in Fig. 9). In the case of the LES domains D06 and D07, the road NO emissions can be seen on the maps; in particular, the 33.3-m LES produces a distribution of NO in which the presence of the streets is clearly visible. Due to the morning transition, the dispersion of the NO is very weak at the early morning. Contrary to the situation with NO, NO$_2$ is not directly emitted at the surface, but it is rapidly produced by the conversion from NO. Therefore, the distribution of NO$_2$ is not as concentrated along the traffic routes as that of NO, but is more influenced by the air flows. In the early morning, the boundary




layer transits from stable to unstable conditions, and the combined effect of shear and surface heating plays an increasingly important role in turbulence generation (Beare, 2008), causing the flow to evolve into horizontal rolls oriented along the mean wind direction (southwesterly in this case, see Fig. 9h, l). As a result, a clear strip-like structure associated with convective rolls is visible in the $NO_2$ distribution patterns, especially in the LES model with 33.3-m resolution (see Fig. 9j). Similarly, the

425    $O_3$ distribution is influenced by the air flow, showing an opposite feature compared to the pattern of $NO_2$ since surface $O_3$ is strongly titrated by NO. To some extent, the spatial distribution of $NO_2$ and $O_3$ can be explained by the pattern of vertical wind speed (Fig. 9h, l). The narrow strip-like updrafts in the southwest direction transport the high surface NO upwards to consume $O_3$, resulting in narrow regions of high (low) concentrations of $NO_2$ ($O_3$) above the near surface layer.

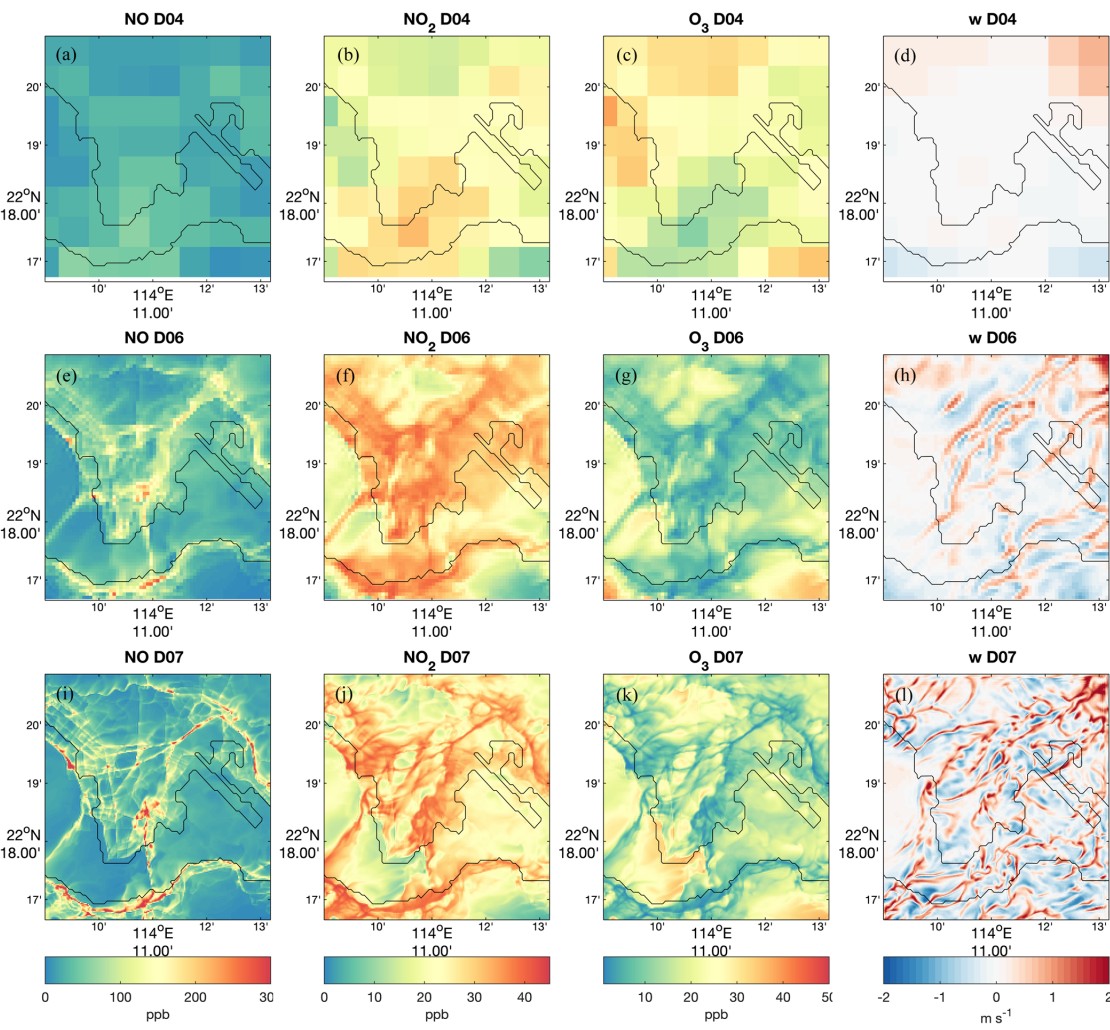

**Figure 10.** Same as Fig. 9, but at 13:00 LT.

430

At 13:00 LT, the NO concentrations are higher than those at 6:00 LT, due to increased traffic emissions during daytime (see Fig. 10). Moreover, the NO emitted on the streets exhibits an obvious diffusion phenomenon in the LES domains at noontime





in response to the enhanced turbulent mixing and the higher-resolution emission distribution applied to the model. The changes with time in $NO_2$ and $O_3$ concentrations at different locations and with different model resolutions are partially due to the complex photochemical reactions. After sunrise, convective instability in the PBL gradually increases, which causes the stronger turbulent mixing and the structure of turbulent flows changes from roll-like to cell-like patterns, yet disturbed by the presence of terrain and land cover heterogeneities. This can be seen from the cellular structures of the vertical wind speeds, and specifically from the comparable intensity of stronger narrow updrafts and wide downdrafts in the more convective boundary layer (see Fig. 10l). Such stronger mixing results in clearly cellular structures in the $NO_2$ and $O_3$ concentration distribution at noontime in the LES D07 domain, but not in the mesoscale domain D04 and in the coarse resolution LES D06 domain.

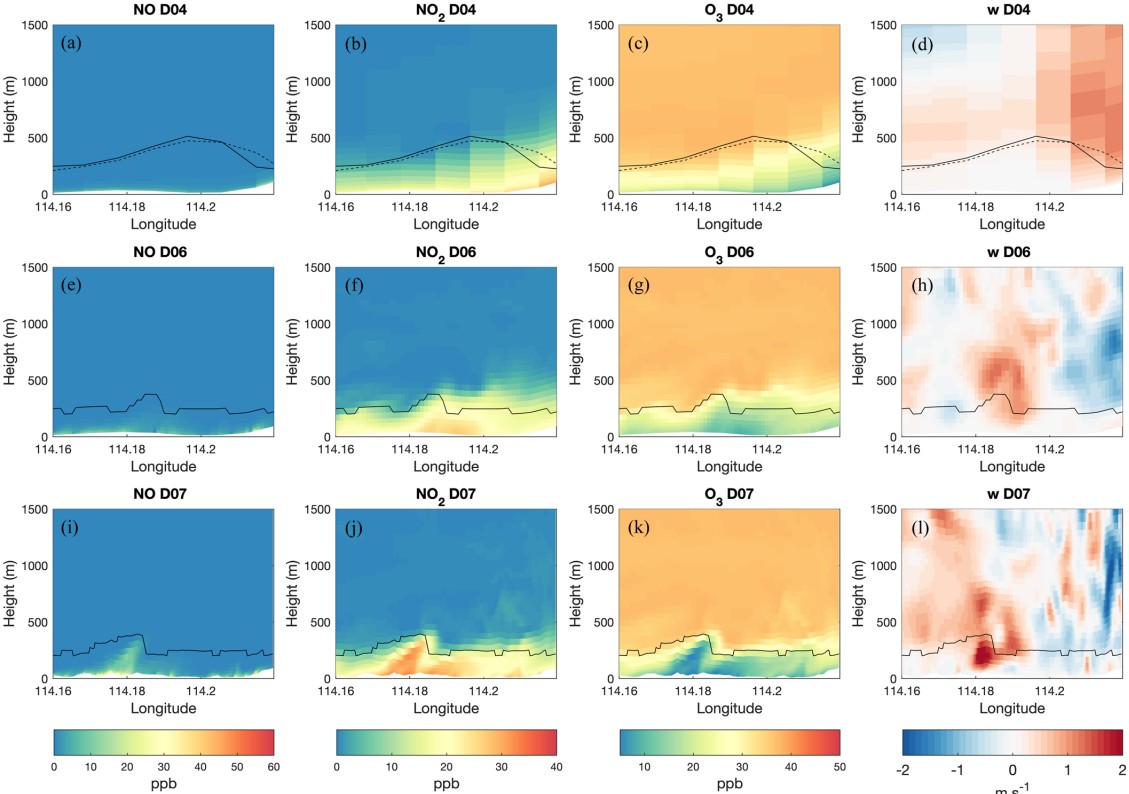

**Figure 11.** Vertical distribution of simulated NO (first column), $NO_2$ (second column), $O_3$ (third column) mixing ratios (ppbv), and vertical wind velocity (w; fourth column) along a busy east-west road near Mong Kok station (Argyle Street, see Fig. 3b) at 6:00 LT for the D04 (first row), D06 (second row), and D07 (third row) cases. The black solid lines represent the calculated PBLH, and the black dash line the PBLH from the model output for the mesoscale domain, which is calculated in the YSU PBL scheme.

The vertical distributions of the chemical species along the busy east-west Argyle Street (see Fig. 3b) in the vicinity of the Mong Kok station are shown in Fig. 11 (6:00 LT) and Fig. 12 (13:00 LT). In the early morning, the NO concentration is low, and the values are close to zero above the surface, implying weak surface emissions together with weak vertical mixing. The surface $NO_2$ concentration becomes higher than that of NO, because of the higher $O_3$ concentrations originating in the upper



layers. The low O₃ values at the surface are caused by the depletion of this molecule by NO. The vertical rate of mixing in the mesoscale model is parameterized in the PBL, while it is directly calculated in the LES simulations. During the morning period, although the selected PBL scheme and the LES formulation produce similar PBL heights, the LES model produces a more detailed structure that better describes the vertical transport of the pollutants. At the noon time, the PBL height increases

to about 1000 m due to the strong convective instability, which also can be seen from the sounding measurements in Fig. 5. The surface NO concentrations are high along the road associated with the traffic emissions, and corresponding high NO₂ and low O₃ concentration values. The vertical mixing of the chemical species at 13:00 LT is significantly stronger than in the morning, because the PBL has become more convective. The vertical transport of the pollutants is higher in the mesoscale simulation than the LESs, especially in the case of NO₂. This suggests that the YSU scheme produces strong vertical diffusion.

Compared with the coarse-resolution LES case (D06), the width of the high NO₂ and low O₃ concentration patterns within the PBL in the high-resolution (D07) LES simulation is significantly reduced (about one to several hundred meters), and the width of the low NO₂ and high O₃ patterns also decreases to the range of several hundred meters to one kilometer. This is related to the thermal structure characterized by narrow updrafts and wide downdrafts during the well-developed phase of the turbulent PBL (see Fig. 12l), which is subject to under-resolved convection effects widening these thermals at coarser grid spacings.

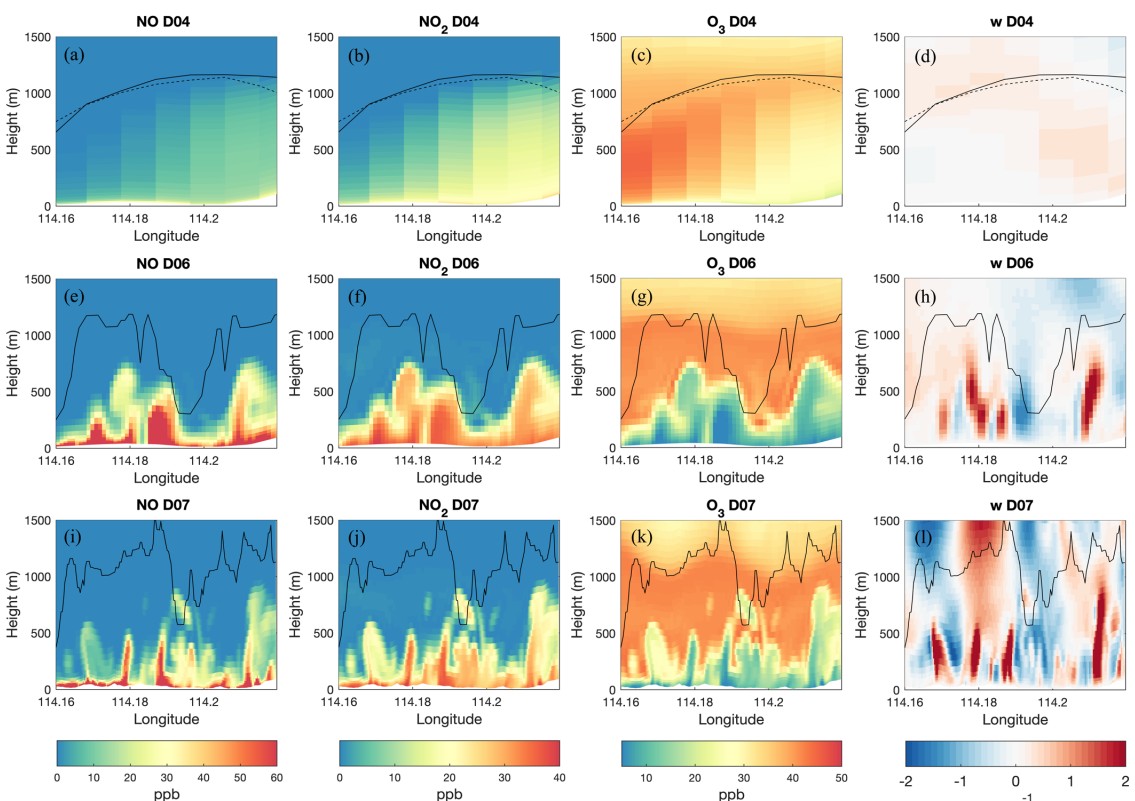


**Figure 12.** Same as Fig. 11, but at 13:00 LT.





## 4.5 Development of vertical mixing of pollutants over a diurnal cycle

As shown in the section 4.4, the vertical structure of the pollutants significantly varies with different model resolutions. To further evaluate the model behavior regarding the overall vertical mixing, the horizontally averaged (over the subregion plotted
in Fig. 9-10) mean profiles of the chemical species at different times of the day are shown in Fig. 13. The PBLH is only about 230 m in the early morning when the boundary layer is still stable. With the intensifying of the solar radiation, the PBLH starts to rise with time and reaches about 400 m at 8:00 LT. The PBLH continues to increase during the morning and reaches the highest height at noon time. The mean PBLH derived for the selected area D04 is 1088 m at 13:00 LT, which is similar to the height calculated for the sounding profile at King's Park station. However, the mean PBLH is 768 m in the D06 LES simulation,
which is considerably lower. This is possibly due to the local dynamics as seen in Fig. 12. The D07 LES is producing a PBLH (977 m) that is higher than the D06 LES simulation, but still 100 m lower than the mesoscale simulation. This indicates that the D06 LES underestimates the strength of the convective motions as it does not resolve turbulent eddies as accurately as the high-resolution LES (D07). In the afternoon, as the surface heat becomes weaker, the PBLH decreases gradually. After sunset, the boundary layer becomes stable again, and the PBLHs declines rapidly to 393 m, 284 m, and 280 m respectively for the
three domains.

The simulated NO concentrations are highest near surface and decrease rapidly against altitude. The mean NO concentrations are between 20 ppb and 30 ppb at the lowest layer of the model in the early morning and decline by a factor of 10 to 1 – 3 ppb at the PBLH. At about 200 m, the high-resolution LES (D07) provides lower NO concentrations relative to the low-resolution LES (D06) and the mesoscale simulation (D04). In other words, the vertical transport of NO is weakest in D07
in the early morning. As the strength of traffic pollution increases during the morning rush hour, the surface NO concentrations increase rapidly to 70 – 90 ppb in the three simulations and reach values that are about three times higher than those at 6:00 LT. The coarse resolution model produces surface NO values that are larger than in the high resolution LESs; however, the NO concentrations at the PBLH are about 10 ppb in all three simulations with higher PBLH derived in the LES simulations. This implies the faster development of the boundary layer and the stronger vertical mixing in the high-resolution LES. The
surface concentration of NO decreases around noon due to the reduction of the road emissions after the rush hour. In contrast with the morning profiles, the D07 LES simulation exhibits high surface NO concentrations. Indeed, values derived in the D07case are about twice as high as the values produced by the mesoscale simulations. However, the vertical mixing of NO above 500 m is stronger in the mesoscale simulation in which the PBL is parameterized. This is related to the higher PBLH produced in the D04 simulation for the convective dominated noon condition. The strong vertical transport extends into the
afternoon, and the profiles at 17:00 LT are similar to those at 13:00 LT. The NO concentrations above 500 m start to decrease



in the evening as the PBL height decreases; however, the surface NO concentration rises again during the evening rush hour at 19:00. It also shows in the high-resolution LES simulations significantly higher NO concentrations in the residual layer.

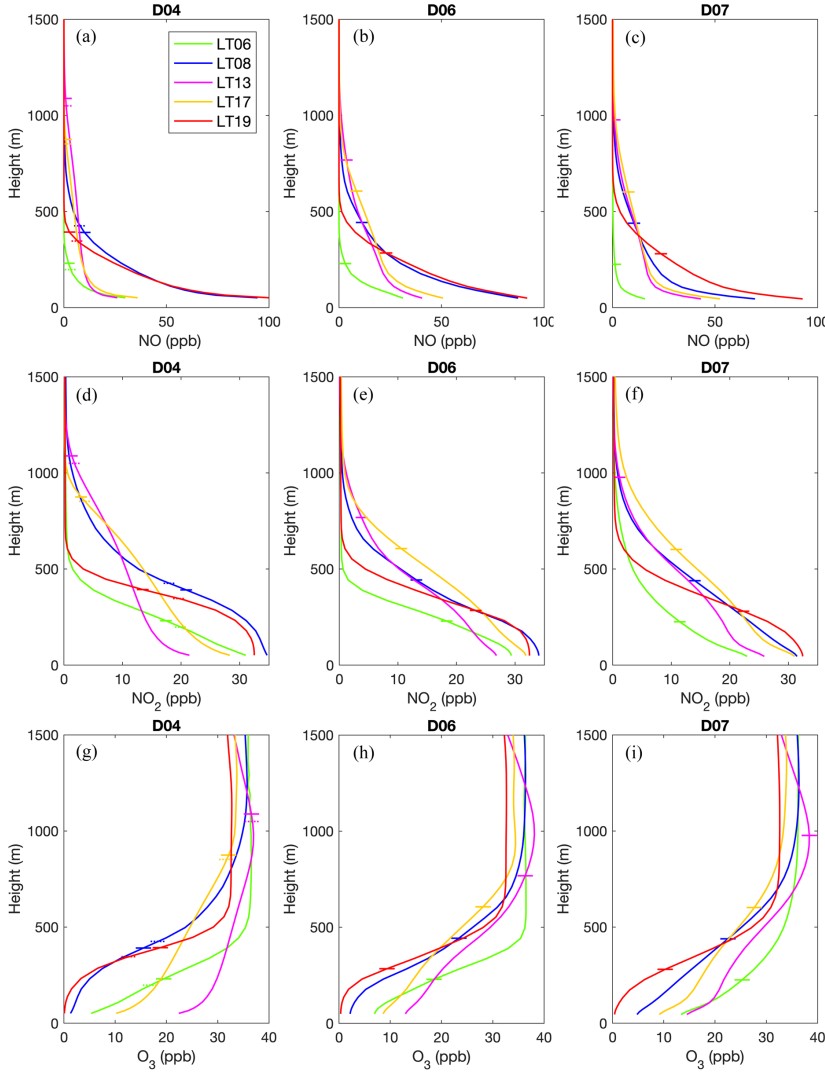

**Figure 13.** Horizontally averaged hourly mean profiles for the mixing ratio (ppbv) of chemical species (NO, $NO_2$ and $O_3$) over plotted region in Fig. 9–10 over the diurnal cycle on August 1, 2018. Different colors represent different times of the day: green – 6:00 LT, blue – 8:00 LT, magenta – 13:00 LT, yellow – 17:00 LT, red – 19:00 LT. Left column: simulation for D04 with resolution of 900 m; middle column: simulation for D06 with resolution of 100 m; right column: simulation from D07 with resolution of 33.3 m.

The patterns that characterize the $NO_2$ mean profiles are different from those of NO. The peak values of $NO_2$ are also located at the surface, due to the large amount of NO, but the decay rate against altitude is slower than that of NO. At 6:00 LT, the surface $NO_2$ concentrations are about 30 ppb in the D04 and D06 simulations and approximately 25 ppb in the D07 LES case. The corresponding concentrations at PBLH are 17 ppb, 18 ppb, and 12 ppb for the three simulations, equivalent to approximately 50% of the surface level concentrations. Compared to the other two simulations, the D07 LES exhibits higher



NO$_2$ concentrations above the PBLH, implying strong conversion of NO to NO$_2$. With the increase of traffic emissions in the morning rush hour at 8:00 LT, the NO$_2$ concentrations did not rise as much as NO, possibly due to the onset of the photolysis.

In the morning, the NO$_2$ concentrations in the upper boundary further increase with the development of the convection in the PBL. Similar to the situation with NO, the mesoscale model generates stronger vertical mixing of NO$_2$ at noon compared to the LES simulations. The situation is opposite in the afternoon: the NO$_2$ concentrations near the PBLH at 17:00 LT became lower than the noon time in the mesoscale simulation, while they continue to increase in the D07 LES. This reveals that the decay of convective condition occurs later in the LES than in the mesoscale model. During the evening hours, the air masses

with high NO$_2$ values move down with the height of the PBL, and, at the same time, the surface NO$_2$ concentrations increase due to the continuous surface emissions and the reduction in the photolysis rates.

The O$_3$ concentration profiles generally exhibit opposite patterns relative to NO$_2$, in response to the titration by NO. The O$_3$ concentrations above the boundary layer are mostly influenced by the background levels, which are between 30 ppb and 40 ppb. In the early morning the surface O$_3$ concentration is about 5 ppb in the D04 case, 7 ppb in the D06 case, and 12 ppb

in the D07 case, respectively; these values are related to the NO concentrations in the different simulations. The surface O$_3$ is reduced at 8:00 LT by the depletion of NO that is released by traffic emissions during the rush hour. The O$_3$ levels in the mixed layer are replenished through subsidence from the upper layers during the day, which results in the rise of O$_3$ during the afternoon. In the evening, O$_3$ concentrations decreases again because of the slower downward transport and the titration by the NO emitted from traffic roads.

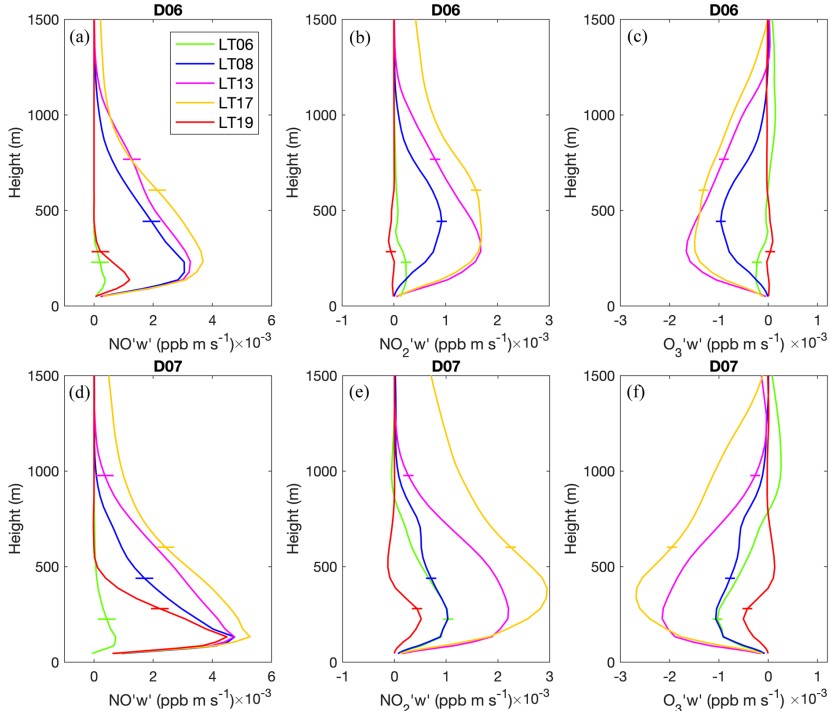

**Figure 14.** Horizontally averaged hourly mean profiles of the vertical fluxes of NO (left), NO$_2$ (middle), and O$_3$ (right) in selected regions highlighting the diurnal cycle on August 1, 2018.




The vertical fluxes of the chemical species are calculated for the LES simulations with different horizontal resolutions. The horizontally averaged flux profiles over the selected region at different times are shown in Fig. 14. For NO, the vertical fluxes

are positive, indicating net upward transport during the entire diurnal cycle. The peak of the NO fluxes appears to be located between the heights of 100 m and 200 m, in a layer where the turbulent motion is strong. The vertical fluxes of NO are smallest at 6:00 LT, when the boundary is stable, and then increases substantially with the development of the PBL after sunrise and the increase in traffic emissions around 8:00 LT. This enhancement of the NO fluxes continues into the afternoon as the convection in the PBL keeps growing, and the peaks of the flux profiles are gradually lifted up. The upward transport becomes

weaker again in the evening when the boundary layer becomes stable. For $NO_2$, the vertical fluxes are also essentially positive, except right above the PBL in the evening. The peaks in the $NO_2$ upward fluxes are higher than those in NO, because of the smaller gradient in the concentration profiles (Fig. 13). The $NO_2$ fluxes above the PBLH are larger than the fluxes of NO, because of the atmospheric photochemical production of $NO_2$. For $O_3$, the vertical fluxes are generally negative, indicating the downward transport from the upper atmospheric layers, and the pattern of the $O_3$ fluxes is opposite to that of $NO_2$. As expected,

the 33.3-m LES produces larger resolved vertical fluxes than the 100-m LES, while the patterns are similar for the two cases. This indicates that the high-resolution LES better represents the turbulent motions, and generates stronger vertical mixing in the convective boundary layer. This is consistent with the analysis of concentration profiles.

## 5 Conclusions

In this paper, we present high-resolution air quality simulations using the WRF model coupled with a LES module and with

an interactive (online) chemical mechanism. This directly coupled mesoscale-to-LES model within the WRF frame ensures the consistency between the physical and chemical processes among scales. In addition to the regular emission inventories used by the coarse models, line sources of the traffic emissions are incorporated to the high LES resolutions. The performance of the WRF-LES-Chem model is evaluated in the megacity Hong Kong, which exposes multi-type chemical sources and the complex topography. The multi-resolution simulations from mesoscale to LES scale are evaluated by comparing to ozone

sounding profiles and measurements from the surface monitoring network. The spatial distributions of the chemical species represented by simulations with different resolutions are analyzed to demonstrate the capability of the model to reproduce the transformation of the pollutants in turbulent flows. The patterns of the vertical mixing of the chemical species are also presented to evaluate the model behavior during the diurnal evolution of the boundary layer.

The comparison with the ozone sounding measurements indicates that the coarser resolution model fails to reproduce the

vertical structure of $O_3$ concentrations, while the higher resolution LES improves the performance near the surface with a better representation of the local emissions. However, large discrepancies still exist in the middle of the boundary layer, which is possibly due to the biases in the wind prediction.

The simulated $NO_X$ and $O_3$ concentrations were validated relative to the Hong Kong EPD surface observations at 16 stations, including 3 roadside stations and 13 general stations. Both mesoscale and LES simulations successfully reproduce the mean



concentrations and the diurnal variations at the general stations, especially at the stations in the suburban areas of the city. The improvement provided by the LES formulation is limited when considering the background concentrations of the pollutants. At the roadside stations, the mesoscale simulation largely underestimates the $NO_X$ concentrations and overestimates $O_3$ due to the coarse representation of the traffic emissions. The LES simulations improve the agreements with the measurements, especially the simulation performed with a 33.3-m grid spacing. However, a statistical analysis of the results shows that an

increase in the LES resolution from 100 m to 33.3 m does not always improve the model results relative to local measurements.

The LES simulations provide more detailed structures of the spatial distributions of the chemical species relative to the mesoscale simulations, because the LES formulation resolves the large turbulent eddies. This is more clearly illustrated when the PBL is well developed at the noon time. The 33.3-m LES provides more vigorous turbulence than the 100-m LES, and thus stronger vertical mixing of the pollutants. The mean profiles of the chemical species at different times are controlled by

the diurnal cycle of the surface emissions, the photolysis rate, and the development of the boundary layer. The simulations with different resolutions show similar trends in the diurnal evolution of the profiles of the chemical species in the boundary layer with some offset time.

The evaluation of the coupled WRF-LES-Chem model shows that the LES simulations provide more detailed distributions of the pollutants resulting from the more detailed representation of the emissions and the explicit representation of the most

energy-carrying turbulence structures. The LES simulations should be further improved with the adoption of real-time traffic emissions rather than the yearly-averaged values used in this work. The multi-scale LES simulation provides encouraging results for future accurate forecasts of air quality in large cities with heavy pollution. Such approach has great potential for operational forecasting in the future with the availability of increasing of the computing resources that enable representation of resolved building effects at grid spacings smaller than 10 m.

*Code and data availability.* The WRF model is public available at https://www2.mmm.ucar.edu/wrf/users/. The air quality data at surface stations are public available on the Hong Kong EPD's website: https://cd.epic.epd.gov.hk/EPICDI/air/station/?lang=en. The ozone sounding data are downloaded from the World Ozone and Ultraviolet Radiation Data Centre: https://woudc.org/home.php.

*Author contributions.* Conceptualization, GPB and YW; methodology, YW and YFM; software, YW, YFM, DME, and JD;
validation, YW; formal analysis, YW; investigation, YW; resources, RCWT; data curation, YW; visualization, YW and YFM; writing—original draft preparation, YW and YFM; writing—review and editing, GPB, CWYL, PL, DME, and CHL; supervision, GPB and TW; project administration, TW; funding acquisition, TW. All authors have read and agreed to the published version of the manuscript.

*Competing interests.* The authors declare that they have no conflict of interest.



*Acknowledgments.*This research is supported by the Hong Kong Research Grants Council (grant no. T24-504/17-N). YFM
contribution to this work is supported by the National Natural Science Foundation of China (NSFC award No.: 42075078).
The National Center for Atmospheric Research is sponsored by the US National Science Foundation. We would like to
acknowledge the high-performance computing support from NCAR Cheyenne. The high-resolution emission data for Hong
Kong is provided by the Hong Kong Environmental Protection Department for purpose of scientific research.

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
