# Peer review of "Coupled mesoscale-LES modelling of air quality in a polluted city using WRF-LES-Chem"

_EGUsphere, 2022_

## Author Comment (AC1)

We thank the referee#1 for taking the time to read the manuscript and offer helpful comments and suggestions. We have modified the manuscript according to the referee's comments. The detailed changes can be found in the word-tracking in the manuscript. The point-to-point responses to the referee's comments are listed below. The referee's comment is repeated with our response in bold.

This article coupled mesoscale and large eddy scale models to simulate air quality in a densely populated city, shows the effect of spatial resolution on the model results and identifies the effect of turbulence on atmospheric chemistry. The content of the whole article is integrated and the model built in this study will effectively promote the air quality forecast to reach the large eddy scale. It is suggested to accept this manuscript with a minor revision. But I still have the following suggestions and questions about the article:

1. The simulation period in this article is short. Although it is difficult for CFD and LES models to run a longer simulation, but as a state-of-art study, could the authors try to simulate or give some discussion about the simulation of the pollution processes, for instance, the pollution for several consecutive days and its elimination?

**Response: We agree with the reviewer that a longer simulation period covering a full pollution process could provide more insight on the LES simulations. However, due to the long computing time, we cannot quickly extend the runs in the current paper. Additionally, the simulation date was chosen based on the sample time of the ozone sounding observations (one measurement per week), as well as the convective boundary layer type. So that the pollution event is not considered in the current work. After this general evaluation, we will investigate more on the pollution processes under different weather conditions in the next steps.**

2. How is the urban canopy scheme and its parameters set up in this study? It is suggested to clarify in the method section.

**Response: The urban canopy scheme is not used in this study, so the urban's effects are only reflected through multiple constant surface parameters (e.g., albedo, roughness, heat capacity, thermal conductivity, etc.) combined with the urban fraction in the land-use data. We have clarified it in the method section. One may expect that urban canopy model is important for the simulations in highly urbanized area, since it may improve the accuracy of the surface and boundary layer properties (Chin et al., 2005; Ching 2013). However, there are great uncertainties in the applications of the urban parameters and urban canopy parameterizations. First, different resolutions and urban morphological descriptions may be required for different urban areas to be "fit" for the purpose (e.g., Baklanov et al., 2009; Ching, 2013), because each city has its own unique degree and characteristic of urban metabolism. This requires many tests, validations and adjustments of urban parameters based on target observations. Second, the accuracy of derived urban properties is sensitive to the resolution of land-use data used (e.g.,**

Chin et al., 2005), as well as the definitions and processing methods (Ching, 2013; Cai et al., 2016). Third, urban canopy parameterizations are sensitive to the urban canopy parameters that define the urban morphology (Salamanca et al., 2011). More importantly, there is also great uncertainty in the simulation results when using urban canopy models. Many studies have shown that the model's performance is sensitive to the urban parameters and urban canopy models, different meteorological conditions, and different variables (e.g., Salamanca et al., 2011; Oleson et al., 2008). Therefore, to avoid additional uncertainties caused by the urban parameters, the urban canopy scheme is turned off for both mesoscale and LES runs for consistency. In the next step, we will investigate the impact of urban parameters and the different urban canopy models (single layer model and multi-layer model) on the simulations of both physical and chemical variables.

3. There is no verification result of simulated and observed meteorological data in the article, and it is impossible to explain the phenomenon well in terms of meteorological factors. Will it be added or explained in the supplement?

**Response: In this paper, we compared the simulations with the vertical profiles of the potential temperature, water vapor mixing ratio, wind speed and wind direction measured by the ozone sounding. The result shows that the LES simulations obtained similar meteorological fields to the mesoscale simulations, and confirms that the LES can reasonably capture the boundary layer development. As for the surface stations, there is no co-measured meteorological data at the EPD surface stations. However, Hong Kong Observatory (HKO) operates the standard meteorological observations at separated stations (see Figure R1). We compared the simulated temperature (T), wind speed (WS), and wind direction(WD) with HKO measurements. The time series averaged from the stations covered by D06 and D07 are shown in Figure R2. It shows that the mesoscale and LES simulations obtained similar trends and can generally match the observations, which is consistent with the sounding comparison. The LES shows some improvements in the wind simulations, while the simulated temperature is a bit worse. This meteorological evaluation has been added into the supplement.**

[Figure]

**Figure R1. Map of the HKO stations covering the simulated period. Red circles are the sites with temperature observations; blue crosses are the sites with wind observations.**

[Figure]

**Figure R2. Time series of temperature (T), wind speed (WS), and wind direction (WD) averaged from stations covered by D06 (left) and D07 (right). The black pentagrams are the observations; the circles with different colors are the simulations with different resolutions (green: D04, 900 m; magenta: D06, 100 m; yellow: D07, 33.3 m). Error bars refer to the standard deviations.**

4. The article introduces the simulation results of roadside stations and ordinary stations, but it is not intuitive enough. Could the authors add time series diagrams for comparative analysis?

**Response: We have further categorized the general stations into urban, suburban, and rural stations. The station types are listed in Table 1. To make it clearer in the time series comparison in Figures 7 & 8, we have marked the station types with different colors for the station names.**

5. Regarding the overestimation of NOx simulation and the underestimation of $O_3$ simulation at some sites, could the authors further analyze it from the aspect of VOCs and explain it in combination with the actual industrial distribution?

**Response: Since the VOCs are not measured at the surface stations, it would be difficult to explain the $NO_X$ and $O_3$ mismatches from the aspect of VOCs, because we do not know if the VOCs are right. We think the overestimation of the $NO_X$ at some stations is not related to the industry, because the industries are with a distance to those stations (see Figure 2b). One possible reason for the overestimation of the $NO_X$ is that the road**

emission is overestimated at the surface, because all the roads (including the roads above ground) emissions are added into the first layer and some stations are lower than the overpass. We added this explanation to the revised manuscript.

6. The vertical profiles in Fig 5 do not show a significant difference between mesoscale WRF and LES-WRF. In other words, the simulation accuracy of LES-WRF is not higher enough as we expected. Could the authors further show some comparison of potential temperature, water vapor mixing ratio, wind speed, wind direction, ozone mixing ratio, etc. inside the PBL or city surface layer?

Response: The vertical structure of the meteorological and chemical fields is determined by both the large scale transportation and the local variation. In the mesoscale models, the large scale motions are resolved while the turbulent eddies are parameterized. In the LES, the large scale structures are constrained by the mesoscale model, and the turbulent mixing is resolved. Therefore, the vertical profiles above the boundary layer are expected to be consistent between the mesoscale and LES simulations. As for the boundary layer, the similarity between the mesoscale and LES does not mean that the accuracy of LES is not enough. It indicates that the YSU scheme and the LES produced similar vertical mixing in this case. We plotted the same figure with Fig. 5 but below 800 m to show clearer comparison in the boundary. It is added into the supplement.

[Figure]

Figure R3. Comparison between ozone sounding measurements and model simulations in the boundary layer at 13:55 LT. The variables are the potential temperature ($\theta$; a), the water vapor mixing ratio ($q$; b), wind speed (WS; c), wind direction (WD; d), and ozone mixing ratio (O3; e). The black lines represent observations; the green lines the simulations from D04 with resolution of 900 m; the magenta lines the simulations from D06 with resolution of 100 m; and the yellow lines the simulations from D07 with resolution of 33.3 m.

7. Could the authors discuss some potential bottlenecks for the use of WRF-LES-Chem in future air quality predictions?

Response: One bottleneck of using WRF-LES-Chem in future air quality prediction is that the original WRF based on terrain-following coordinates with a resolution of more than several ten meters (for high-resolution LES mode) cannot resolve buildings, which is becoming important if the resolution further increases to 10 m or less. To solve it, an alternative meshing technique, which is called immersed boundary method (IBM) is adopted (Lundquist et al., 2010; Lundquist et al., 2012). Another disadvantage is the

**huge amount of computing time, which makes it difficult to apply it in the real-time forecast. This may be improved by accelerating WRF by leveraging GPUs. Some work has been done by different groups, e.g., WRFg, https://wrfg.net/wrfg-description/. With such further developments of the model system, opportunities exist for optimizing the WRF-LES-Chem in future air quality prediction. This has been added in the discussion in the revised manuscript.**

References:

Baklanov, A., Grimmond, C. S. B., Mahura, A., and Athanassiadou, M.: Meteorological and Air Quality Models for Urban Areas, Springer Berlin, Heidelberg, 184 pp., 10.1007/978-3-642-00298-4, 2009.

Cai, M., Ren, C., Xu, Y., Dai, W., and Wang, X. M.: Local Climate Zone Study for Sustainable Megacities Development by Using Improved WUDAPT Methodology – A Case Study in Guangzhou, Procedia Environmental Sciences, 36, 82-89, https://doi.org/10.1016/j.proenv.2016.09.017, 2016.

Chin, H.-N. S., Leach, M. J., Sugiyama, G. A., Leone, J. M., Walker, H., Nasstrom, J. S., and Brown, M. J.: Evaluation of an Urban Canopy Parameterization in a Mesoscale Model Using VTMX and URBAN 2000 Data, Monthly Weather Review, 133, 2043-2068, https://doi.org/10.1175/MWR2962.1, 2005.

Ching, J. K. S.: A perspective on urban canopy layer modeling for weather, climate and air quality applications, Urban Climate, 3, 13-39, 10.1016/j.uclim.2013.02.001, 2013.

Lundquist, K. A., Chow, F. K., and LUNDQUIST, J. K.: An Immersed Boundary Method for the Weather Research and Forecasting Model, Monthly Weather Review, 138, 796-817, 10.1175/2009mwr2990.1, 2010.

Lundquist, K. A., Chow, F. K., and Lundquist, J. K.: An Immersed Boundary Method Enabling Large-Eddy Simulations of Flow over Complex Terrain in the WRF Model, Monthly Weather Review, 140, 3936-3955, https://doi.org/10.1175/MWR-D-11-00311.1, 2012.

Oleson, K. W., Bonan, G. B., Feddema, J., and Vertenstein, M.: An Urban Parameterization for a Global Climate Model. Part II: Sensitivity to Input Parameters and the Simulated Urban Heat Island in Offline Simulations, Journal of Applied Meteorology and Climatology, 47, 1061-1076, https://doi.org/10.1175/2007JAMC1598.1, 2008.

Salamanca, F., Martilli, A., Tewari, M., and Chen, F.: A Study of the Urban Boundary Layer Using Different Urban Parameterizations and High-Resolution Urban Canopy Parameters with WRF, Journal of Applied Meteorology and Climatology, 50, 1107-1128, https://doi.org/10.1175/2010JAMC2538.1, 2011.

---

## Author Comment (AC2)

**Response to Interactive comment from Referee #2**

**We thank the referee#2 for taking the time to read the manuscript and offer helpful comments and suggestions. We have modified the manuscript according to the referee's comments. The detailed changes can be found in the word-tracking in the manuscript. The point-to-point responses to the referee's comments are listed below. The referee's comment is repeated with our response in bold.**

Summary

This paper presented high-resolution air quality simulation model using the WRF model with 7 nesting levels, where the innermost 3 domains are modelled through WRF-LES, using on-line chemistry with the RADM2 mechanism. The model was evaluated in Hong Kong, which has a complex topography and multi-type chemical sources. The LES model performed better than coarser models in terms of reproducing ozone concentration profiles and diurnal variations in mean concentrations at observation stations. However, the model had limitations in reproducing the NOX concentrations and overestimating O3 at roadside stations due to the coarse representation of traffic emissions. Despite this, the Authors asserted the potential of the multiscale approach (using WRF and WRF-LES) for accurate air quality forecasting, as it provided better details in pollutant distributions the explicit representation of energy-carrying turbulence structures.

General comments

Congratulations for undertaking such a challenging problem. As indicated by the authors, even in the presence of highly resolved spatiotemporal dynamics (in this case LES), the characterization of emission sources remains problematic, have reverberated our own research. The manuscript is generally well prepared, albeit there are few technical and rhetorical concerns, listed below, in addition to the following general concerns, which should be adequately addressed prior to publication.

**Response: We thank the reviewer for the general comments on our work. The general conclusion is that the mesoscale models have limitations in reproducing the $NO_X$ and $O_3$ concentrations at the roadside stations due to the coarse representation of the traffic emissions; however, the LES models show some improvements on this, although there are still some mismatches. The disagreements between the LES simulations and the roadside measurements are not due to the coarse spatial representation of the traffic emissions. The original traffic emissions used in this work are in the form of line sources for every road, and they are interpolated to the model grids of each domain. As for the temporal variations, the annual totals were evenly distributed to every day and then disaggregate to hours with a diurnal profile (Fig R1). We have added more explanations in the revised manuscript. We also add the diurnal profiles in the supplement.**

[Figure]

**Figure R1. The diurnal profiles of the road and point emissions. Black dashed line represents the road emissions; colored solid lines represent various point emissions (power plants, industries, crematorium, and tank farms) with different diurnal variations.**

1. As the study concerns a heavily urbanized region, the type of urban canopy model that was used (in the RANS domains) should be indicated. The expectation is that these models, should provide representative surface fluxes in the urban region and allow explicit resolution in the LES domains. It might be worthwhile to provide some form of commentary on this and the impacts of using and not using an urban canopy model.

**Response: In this study, we did not use any urban canopy model. The urban's effects are only reflected through multiple constant surface parameters (e.g., albedo, roughness, heat capacity, thermal conductivity, etc.) combined with urban fractions in land-use data.**

**As you might expect, urban canopy model is very important for the simulation of a heavily urbanized region. Using urban parameters/urban canopy parameterization schemes may improve the accuracy of the model for the surface and boundary layer properties, in particular in wind fields and pollutant distributions (Chin et al., 2005; Ching, 2013). However, there are great uncertainties in the applications of urban parameters and urban canopy parameterizations. First, different resolutions and urban morphological descriptions may be required for different urban areas to be "fit" for the purpose (e.g., Ching, 2013; Baklanov et al., 2009), because each city has its own unique degree and characteristic of urban metabolism. This requires many tests, validations and adjustments of urban parameters based on target observations. Second, the accuracy of derived urban properties is sensitive to the resolution of land-use data used (e.g., Chin et al., 2005), as well as the definitions and processing methods (Cai et al., 2016; Ching, 2013). Third, urban canopy parameterizations are sensitive to the urban canopy parameters that define the urban morphology (Salamanca et al., 2011). More importantly, there is also great uncertainty in the simulation results when using urban canopy models. Many studies have shown that the model's performance is sensitive to**

the urban parameters and urban canopy models, and it also depends on the different meteorological conditions and the different variables (e.g., Oleson et al., 2008; Salamanca et al., 2011).

However, the main purpose of our first step is how to make the online coupling WRF-LES-Chem system work properly and reproduce the reasonable results with high-resolution local pollutant emissions, such as line sources (e.g., vehicle exhaust emissions on the road network, ship emissions) and point sources (factories, power plants, etc.). Therefore, we did not use the urban canopy model in this phase due to the large uncertainties of the acquisition and application of urban canopy models.

Considering both thermal and mechanical aspects of sub-grid building effects in mesoscale/microscale models is important to better predict weather, climate, and air quality in urban areas. In the next step, we will investigate the impact of urban parameters and the different urban canopy models (single layer model and multi-layer model) on the simulations of both physical and chemical variables.

2. My understanding on the emission input data presented in Section 2.2 and Figure 2 is that they are stationary (i.e., time-invariant) boundary conditions throughout the model period. While this has been rightfully pointed out as a deficiency for the study (e.g., on page 13 line 322), it would be more sensible to at least discuss or suggest the possibility of disaggregating these values based on, for instance, sector-relevant diurnal profiles (e.g., GNFR or equivalent)?

Response: The emission input data are not stationary in this study. The gridded emissions from oceanic and residual sources are on hourly basis. The original road emission data are the annual totals with a diurnal profile (Fig. R1). The point sources are the annual totals with corresponding monthly and diurnal profiles. Therefore, the emission input data in the model are all pre-processed to 1-hour basis. We have added more information in the revise manuscript, as well as the diurnal profiles in the supplement.

3. It took me a few readings to understand sections 3.1 and 3.2 correctly. The authors might want to make these two sections more concise and clear. For instance, the authors might want to clearly specify, directly on Table 1, what species are being monitored at each station. In addition, I have to assume that "general" stations measure background concentrations. The authors should also indicate this for completeness, especially when these "general" stations are further categorized (e.g., urban, suburban, or rural background) which should also be indicated.

Response: We have added the used species in Table 1 as the reviewer recommended. The general stations are defined by the EPD, who owned the stations and operated the observations. It is relative to the roadside stations. We separated the general stations to rural, suburban, and urban depending on their locations in the revised manuscript.

4.  In section 4.1, the authors presented the PBLH at the time of the sounding profile (13:55 local time) is made, as a form of model evaluation. Showing the PBLH as a single point measurement (in time and space) is inadequate, in my opinion, because the evolution of the PBLH in the model, and thus the vertical mixing, is not known. The discussion on vertical concentrations in Section 4.5 (and Figures 13 / 14) attempts to bring some comparison PBLH between the RANS and LES domains and use that as the basis to estimate the over- / under-predictiveness in the LES domains, but the temporal relationship of this is all but gone, largely due to how Figures 13 and 14 are presented, which makes the understanding quite difficult. I would suggest showing the diurnal profile of the PBLH in the domains considered, in addition to the vertical profiles, to aid the explanation.

**Response: We agree with the reviewer that comparing the PBLH at a single point cannot give enough information on the boundary layer development and the vertical mixing. However, the section 4.1 aims the compare the model results with the sounding profile, which is at a single time, so that the PBLH is an additional term for interpretation. In the revised manuscript, we kept the calculated PBLH for the profile, and added the evolution of the simulated PBLH (also shown here in Fig R2) in section 4.5 as the reviewer suggested.**

[Figure]

**Figure R2. Horizontally averaged PBLH over plotted region in Fig. 9–10 over the diurnal cycle on August 1, 2018. Different colors of the solid lines represent different model resolutions: green – D04 (900 m), magenta – D06 (100 m), yellow – D07 (33.3 m).**

5.  The explanation (i.e., first appearance) of abbreviated terms should be consistent. Sometimes to the full term is first referred and the corresponding abbreviations provided, while other times it is the other way around. In some instances there are no explanation provided for the abbreviated term. A quick example can be found on the first paragraph on Page 5, and the last paragraph of Page 7. I will let the authors sort this out.

**Response: We have modified the abbreviated terms and their explanations accordingly in the revised manuscript.**

6. The captions for Figures 8, 10, and 12, as well as Table 3 should be expanded. While the current approach it is more succinct, it saves the forgetful reader (like yours truly) from needing to constantly refer to the corresponding captions which, given their length and the size of figure / table, can be very cumbersome. This naturally applies to other similar figure and tables on this manuscript not mentioned in this comment.

**Response: We expanded the captions for the figures as recommended.**

7. Referral of the observational stations should be accompanied by a definite article. For example, "at Causeway Bay station" should be "at the Causeway Bay station".

**Response: We have corrected this in the manuscript.**

Specific comments

1. Page 3 Line 76 : It should be made aware that INIFOR has restricted availability as it derives lateral meteorological profiles from proprietary data (COSMO). Instead, the authors are strongly encouraged to refer to WRF4PALM (Lin et al, GMD 14 2503-2524, 2021) as a generally available method for obtaining mesoscale meteorological and chemical boundary conditions from WRF and WRF-Chem model data, which aligns much more closely with the context of this manuscript.

**Response: Thank the reviewer for the useful comments. We have added the WRF4PALM in the manuscript.**

2. Page 5 Line 143 : A brief technical description on the cell perturbation method of Muñoz-Esparza et al in D05 should be provided, in particular, how the turbulent length scales are modelled / parameterized and how they represent the turbulent spectrum commonly encountered in atmospheric flows, different from other methods (e.g., Xie and Castro, Flow Turbul Combust 81 449-470, 2008; as implemented in Zhong et al, GMD 14 323-336, 2021).

**Response: We have added a short description of the cell perturbation method. Because this work does not concentrate on the cell perturbation method, so the details are not included in the manuscript. However, we answer the reviewer's questions here.**

**The digital filtering method developed by Xie and Castro (2008) and implemented by Zhong et al. (2021) is based on an exponential correlation function and artificially generates a series of 2-D fields correlated both in space and in time, which satisfy the prescribed (target) integral length scales and Reynolds-stress-tensor. Although it is more computationally efficient than other synthetic turbulence generation methods (e.g., 3-D digital filter methods) and it's also promising, these synthetic methods present many disadvantages: (1) the generation of temporal correlation is a critical aspect of such synthesized turbulence methods and also a fundamental problem in developing a**

realistic turbulence field; (2) it requires a priori detailed information of turbulent characteristics (e.g., length scales, turbulence intensities, anisotropy, etc.) and are not always divergence-free; (3) it still requires long fetches for the turbulence to develop; (4) it developed for channel-flow applications and neutral stratification (Muñoz-Esparza and Kosović, 2018; Muñoz-Esparza et al., 2014; 2015; Mazzaro et al., 2019). For atmospheric flows, the atmospheric stability is not steady and evolves in time and space, and it has strong impact on length scales, turbulence intensities and anisotropy of the flow, as well as on the mean velocity and scalar distributions (Muñoz-Esparza et al., 2014). Therefore, such velocity-perturbation-based synthetic turbulence method is not very suitable for the real atmospheric environment simulations.

Cell-perturbation method (CPM) based on potential temperature (its vertical gradient can characterize atmospheric stability) perturbation mainly provides a mechanism to accelerate the transition from a mean turbulent flow towards fully developed turbulence, rather than imposing a developed turbulent field at the inflow planes as pursued by synthetic-type methods. CPM applied at wavelengths within the inertial range of 3-D turbulence provides nearly equivalent quasi-equilibrium levels of resolved turbulent kinetic energy (TKE) and Reynolds-shear stress, with a slight increase in the initial energy generated near the inflow boundaries and the required fetch to reach the quasi-equilibrium solution for cell sizes near the low-wavenumber end of the inertial range (Muñoz-Esparza et al., 2015). This method can quickly establish the entire 3-D turbulence spectrum controlled by the optimum Eckert number, including production and inertial range scales, and develop realistic turbulence consistent with reference calculations using periodic transverse boundary conditions.

Mazzaro et al. (2019), Muñoz-Esparza and Kosović (2018), and Muñoz-Esparza et al. (2014; 2015) compared different turbulence generators (including Xie and Castro, 2008) and proved that CPM is suitable for a wide range of ground winds and is not limited by specific mesh resolutions, as well as it's the simplest and most efficient of the best performing methods for the real atmospheric flows, including neutral, stable and unstable conditions.

3. Page 5 Line 144 : For the reader's benefit, the authors might want to elaborate on what "too coarse" means.

Response: The horizontal resolution of 300 m is considered "too coarse" for LES because it can theoretically only resolve the turbulent eddies with length scale above 600 m, and the most turbulent motions cannot be resolved. However, in fact the effective resolution (i.e., the minimum wavelength correctly seen by the model) of the model is always larger than the grid size $\Delta x$, and is typically around 5~6$\Delta x$ (Lac et al., 2018), i.e., the scales from which the model departs from the theoretical slope of -5/3. In this way, the 300-m resolution model can only correctly simulate the flow with wavelengths above 1.5 km. Therefore, a resolution of 300 m is too coarse for the LES model in our case study, which has a PBLH around 1 km. However, to avoid misleading, we modified it to "relative coarse" in the revised manuscript.

4. Page 5 Line 147 : Is one hour of spin-up time sufficient for the LES domains? If this is solely due to constraints on computational resource, and are relying on the spun-up properties from the RANS domains (since they were set to 42 hours), the authors should indicate this explicitly. Otherwise, some substantial form of justification would be necessary.

**Response: The physics and chemistry have reached equilibrium after the spin-up of the mesoscale domains (45 hours), so we only consider a short spin-up time for the LES to let the turbulence develop. Due to the effects of complex terrain and daytime surface heating, as well as the application of the cell perturbation method, the turbulence can actually be well-developed within 20~30 minutes during the daytime (Muñoz-Esparza and Kosović, 2018). In our cases, the turbulence in the whole domain is fully developed within one hour. We have clarified it in the revised manuscript.**

5. Page 7 Figure 3 : Station names on both figures are very hard to see (and I already zoomed it in to 250%). Consider the following measures: 1) apply a background color to each name label, preferably white, and introduce a transparency in the background. 2) use lines and arrows to provide additional spacing between closely clustered stations. 3) Remove lat / long indicators on both figures to maximize figure real estate (as the coordinates are already indicated in Table 1 it is not necessary to state them again graphically).

**Response: We have modified the figure with larger and bold text for the stations.**

6. Page 12 Line 298 : If the NOx emission dataset were, indeed, time-invariant (see corresponding general comment), then this statement of overestimated road emissions during rush hours cannot be true, as the diurnal disaggregation will very likely result in a higher emission than presented during this period. This, in turn, leaves the only interpretation, that the NOx emissions are in fact overestimated all day long, which is not the case here. Please revise this statement accordingly. Also see general comment on PBLH for further information.

**Response: The emission input is not time-invariant. See the response for the general comments above.**

7. Page 12 Line 302 : Now the authors refers to a suburban station. Are they officially classified as such? Please refer to general comment on Section 3.1/3.2 for further details to ensure consistency.

**Response: We have separated the general stations to rural, suburban, and urban stations based on their locations.**

8. Page 27 Line 567 : As a rhetorical question, what exactly are "large" turbulent eddies?

**Response: Here the large eddies refer to the eddies resolved by the large-eddy simulation. The large turbulent eddies mean the largest and most energetic eddies, which can be resolved by the LES model. These eddies can have a wide range of scales from tens meters to several kilometers depending on the specific application of the simulation. Compared to the sub-grid scale, all the eddies resolved by LES model can be seen as large-scale eddies because the effective resolution of LES model is typically around 5~6Δx (Lac et al., 2018), while the Komogorov microscale (i.e., the smallest scales in turbulent flow) for real atmospheric boundary layer flow is typically on the order of millimeters. We added the description of large eddies in the introduction, where intrudes LES, in the revised manuscript.**

Rhetorical remarks

1. Page 5 Line 125 : Did the authors mean to write "IGBP-MODIS" or "IGBP MODIS"? If the former is intended the abbreviation should be explained together so that it does not look like a typographical error. Also see general comment on abbreviations.

   **Response: We modified it in the manuscript to avoid misunderstanding.**

2. Page 7 Line 189 : "heighs" > "heights"

   **Response: It is corrected.**

3. Page 7 Line 177 : "Megan" > "MEGAN"

   **Response: It is corrected.**

4. Page 9 Line 221 : Would it not be easier to explain what theta_vs is directly, instead of explaining first theta_v, and then the subscript s, since the subscript s only exists in conjunction with theta_v?

   **Response: We agree with the reviewer, and have modified it.**

5. Page 11 Line 264 : "carbonyls" > "RCO" (to be consistent with RH and ROx in the other reactions.)

   **Response: We modified it as suggested.**

6. Page 12 Lines 277-279 : The sentence with the with the multiple slashed adjective choices becomes too confusing to read. Consider rewriting to something like this: "In addition, the air parcels with low O3 values are clearly transported to higher attitudes by the updrafts produced by [...]. On the other hand, the downdraft produced by [...] transport the air parcels with high O3 values to the lower attitudes."

   **Response: We have modified it in the revised manuscript.**

7. Page 13 Lines 320-321 : Based on the discussion, I think the authors mean to write ".. high-resolution model does not necessary provide better predictions …" as opposed to "… much better predictions".

**Response: It is corrected.**

**References:**

Baklanov, A., Grimmond, C. S. B., Mahura, A., and Athanassiadou, M.: Meteorological and Air Quality Models for Urban Areas, Springer Berlin, Heidelberg, 184 pp., 10.1007/978-3-642-00298-4, 2009.

Cai, M., Ren, C., Xu, Y., Dai, W., and Wang, X. M.: Local Climate Zone Study for Sustainable Megacities Development by Using Improved WUDAPT Methodology – A Case Study in Guangzhou, Procedia Environmental Sciences, 36, 82-89, https://doi.org/10.1016/j.proenv.2016.09.017, 2016.

Chin, H.-N. S., Leach, M. J., Sugiyama, G. A., Leone, J. M., Walker, H., Nasstrom, J. S., and Brown, M. J.: Evaluation of an Urban Canopy Parameterization in a Mesoscale Model Using VTMX and URBAN 2000 Data, Monthly Weather Review, 133, 2043-2068, https://doi.org/10.1175/MWR2962.1, 2005.

Ching, J. K. S.: A perspective on urban canopy layer modeling for weather, climate and air quality applications, Urban Climate, 3, 13-39, 10.1016/j.uclim.2013.02.001, 2013.

Lac, C., Chaboureau, J. P., Masson, V., Pinty, J. P., Tulet, P., Escobar, J., Leriche, M., Barthe, C., Aouizerats, B., Augros, C., Aumond, P., Auguste, F., Bechtold, P., Berthet, S., Bielli, S., Bosseur, F., Caumont, O., Cohard, J. M., Colin, J., Couvreux, F., Cuxart, J., Delautier, G., Dauhut, T., Ducrocq, V., Filippi, J. B., Gazen, D., Geoffroy, O., Gheusi, F., Honnert, R., Lafore, J. P., Lebeaupin Brossier, C., Libois, Q., Lunet, T., Mari, C., Maric, T., Mascart, P., Mogé, M., Molinié, G., Nuissier, O., Pantillon, F., Peyrillé, P., Pergaud, J., Perraud, E., Pianezze, J., Redelsperger, J. L., Ricard, D., Richard, E., Riette, S., Rodier, Q., Schoetter, R., Seyfried, L., Stein, J., Suhre, K., Taufour, M., Thouron, O., Turner, S., Verrelle, A., Vié, B., Visentin, F., Vionnet, V., and Wautelet, P.: Overview of the Meso-NH model version 5.4 and its applications, Geosci. Model Dev., 11, 1929-1969, 10.5194/gmd-11-1929-2018, 2018.

Mazzaro, L. J., Koo, E., Muñoz-Esparza, D., Lundquist, J. K., and Linn, R. R.: Random Force Perturbations: A New Extension of the Cell Perturbation Method for Turbulence Generation in Multiscale Atmospheric Boundary Layer Simulations, Journal of Advances in Modeling Earth Systems, 11, 2311-2329, https://doi.org/10.1029/2019MS001608, 2019.

Muñoz-Esparza, D. and Kosović, B.: Generation of Inflow Turbulence in Large-Eddy Simulations of Nonneutral Atmospheric Boundary Layers with the Cell Perturbation Method, Monthly Weather Review, 146, 1889-1909, 10.1175/mwr-d-18-0077.1, 2018.

Muñoz-Esparza, D., Kosović, B., Mirocha, J., and van Beeck, J.: Bridging the Transition from Mesoscale to Microscale Turbulence in Numerical Weather Prediction Models, Boundary-Layer Meteorology, 153, 409-440, 10.1007/s10546-014-9956-9, 2014.

Muñoz-Esparza, D., Kosović, B., van Beeck, J., and Mirocha, J.: A stochastic perturbation method to generate inflow turbulence in large-eddy simulation models: Application to neutrally stratified atmospheric boundary layers, Physics of Fluids, 27, 10.1063/1.4913572, 2015.

Oleson, K. W., Bonan, G. B., Feddema, J., and Vertenstein, M.: An Urban Parameterization for a Global Climate Model. Part II: Sensitivity to Input Parameters and the Simulated Urban Heat Island in Offline Simulations, Journal of Applied Meteorology and Climatology, 47, 1061-1076, https://doi.org/10.1175/2007JAMC1598.1, 2008.

Salamanca, F., Martilli, A., Tewari, M., and Chen, F.: A Study of the Urban Boundary Layer Using Different Urban Parameterizations and High-Resolution Urban Canopy Parameters with WRF, Journal of Applied Meteorology and Climatology, 50, 1107-1128, https://doi.org/10.1175/2010JAMC2538.1, 2011.

Xie, Z.-T. and Castro, I. P.: Efficient Generation of Inflow Conditions for Large Eddy Simulation of Street-Scale Flows, Flow, Turbulence and Combustion, 81, 449-470, 10.1007/s10494-008-9151-5, 2008.

Zhong, J., Cai, X., and Xie, Z. T.: Implementation of a synthetic inflow turbulence generator in idealised WRF v3.6.1 large eddy simulations under neutral atmospheric conditions, Geosci. Model Dev., 14, 323-336, 10.5194/gmd-14-323-2021, 2021.

---

## Author Comment (AC3)

**Coupled mesoscale-LES modelling of air quality in a polluted city using WRF-LES-Chem**

Yuting Wang1, Yong-Feng Ma2, Domingo Muñoz-Esparza3, Jianing Dai4, Cathy W. Y. Li4, Pablo Lichtig4, Roy C.W. Tsang5, Chun-Ho Liu6, Tao Wang1, and Guy P. Brasseur1,4,7

[revised manuscript text omitted]

---

## Author Response (AR2)

**Responses to Editor's comments**

**We thank the editor for reviewing the revised manuscript and providing comments for corrections. The point to point responses are listed below:**

In you supplementary figure S1: Please denote which color represents what.

**Response: We have added a legend in the Figure S1 to denote the different diurnal profiles.**

[Figure]

**Figure S1.** The diurnal profiles of the road and point emissions. Black dashed line represents the road emissions; colored solid lines represent various point emissions (power plants, industries, crematorium, and tank farms) with different diurnal variations.

Throughout the manuscript: I am confused by your use of PBLH and PBL. Sometimes you seems to use them synonymous, e.g. P29 line 663

**Response: We defined PBL in p2 line37, which is referred to as planetary boundary layer, and PBLH in p9 line 227, which is the height of the PBL. They are not the same, because PBL is used when we talk about the whole boundary layer and PBLH is referred to as the height the PBL can reach (which is a number). PBLH represents the development of the PBL. Therefore, this two terms cannot be combined in the paper. However, as the editor mentioned in P29 line 663, "above the layer" and "above the height of the layer" is the same, so we have modified it to "above the PBL" for consistency.**